# APPLYING SECOND ORDER OPTIMIZATION TO DEEP TRANSFORMERS WITH PARAMETER-EFFICIENT TUNING

## ABSTRACT

Despite the theoretical superiority in convergence rate, second-order optimizers are generally not among the top choices for training large-scale neural networks due to their high computation and memory cost. Nevertheless, introduced in recent progress of parameter-efficient tuning is a new paradigm that large-scale pre-trained models (PTMs) can be adapted to specific tasks by optimizing a tiny proportion of parameters, which might hopefully change the game. We associate this new paradigm with the computational tractability of second-order optimizers and succeed in applying them to large PTMs from hundreds of millions of parameters to billions in scale. Beyond verifying their tractability, we further investigate the stability-influencing factors in the optimization process and propose accordingly a Newton-step clipping approach in which we clip the update tensors rather than the gradients. This approach stabilizes the convergence by gating the magnitude of Newton steps along the optimization trajectories through the rugged landscapes of deep transformers. We conduct extensive experiments across different downstream tasks, demonstrating that, when equipped with Newton-step clipping, second-order optimizers, especially Kronecker-factored curvature approximation (K-FAC), can attain comparable and even superior results and faster convergence to those state-of-the-art bars implemented with AdamW. Furthermore, we scale the model up to 3 billion parameters and validate the tractability and effectiveness of our method. This work is not only the first successful application of second-order optimization on such enormous models but also paves the road towards the design and analysis of second-order optimizers for the downstream adaptation of large-scale PTMs.

## 1 INTRODUCTION

Pre-Trained models (Bommasani et al., 2021; Han et al., 2021) (PTMs) based on deep transformers (Vaswani et al., 2017; Devlin et al., 2019; Raffel et al., 2020; Brown et al., 2020) yield remarkable performance on a wide range of tasks thanks to the tremendous capacity brought by numerous parameters. A prevalent paradigm is to firstly pre-train such models on large-scale corpora in a self-supervised manner and then adapt them to specific datasets (typically with supervision). Such adaptations are usually implemented by first-order gradient-based optimization. However, recent applications have witnessed an approaching bottleneck for first-order training – it is neither likely to be faster nor easy to attain higher scores (Pascanu et al., 2013; Goodfellow et al., 2016). On the other hand, second-order optimizers which enjoy better convergence properties might have been an ideal alternative in theory (Nocedal & Wright, 1999; Boyd et al., 2004). They are, nevertheless, almost unattended on the downstream adaptation of PTMs in practice. It is because they require quadratic storage and cubic computation time for each update, which is especially prohibitive given the enormous model scale, even though bunches of their simplified counterparts have been devised (Byrd et al., 1995; Martens & Grosse, 2015; Botev et al., 2017; Anil et al., 2020; Tang et al., 2021).

Hopefully, recent advances in PTMs have brought a slight twist to the situation. Studies show that full-parameter optimization may not be necessary for task-specific adaptations, as updating a small portion (0.05%~1%) of parameters can achieve non-trivial performance in many datasets (Houlsby et al., 2019; Li & Liang, 2021; Lester et al., 2021; Hu et al., 2021). With the new paradigm largely shrinking the trainable parameter space, we find in experiments that second-order optimization is

now tractable on large-scale PTMs with up to billions of parameters. This successful implementation signals that the design and analysis of second-order optimization can start marching from relatively toy or medium models to pre-trained deep transformers that are formidably large in scale. Further triggered is a sequence of interesting research topics including: How tractable are second-order optimizers on deep transformers? Are they capable of converging faster and more steadily and yielding better results? If not, what auxiliary skills are needed? And also, how does the ratio of trainable parameters influence the relative performance of second-order optimization?

In this application-oriented paper, we answer all the aforementioned questions through theoretical justification and experimental verification. Taking K-FAC as the major example of the second-order optimizer, we first experimentally verify that its training time and memory cost on extremely large PTMs are totally affordable. In addition, we point out that clipping the gradients before or after Hessian (or Fisher) preconditioning makes a non-negligible difference. We propose accordingly a Newton-step clipping strategy which is indispensable for second-order training because of its superior stabilizing effect. We then post comprehensive results to illustrate that with the assistance of our Newton-step clipping strategy, second-order optimization outperforms baseline first-order optimzers as well as its non-clipping and traditionally gradient clipping counterparts in terms of both convergence speed and final test scores. Moreover, we scale the model scale up to 3 billion parameters to evaluate the tractability and effectiveness. Observations of the relation between tunable ratio and convergence speed are also presented subsequently.

## 2 SECOND-ORDER OPTIMIZERS FOR LARGE-SCALE TRAINING

Following the introductory part, we recap the essential background of second-order optimization. While first-order optimizers such as stochastic gradient descent (SGD) and Adam (Kingma & Ba, 2014) have been more than ubiquitous in deep learning, second-order optimizers remain relatively under-explored in this field. Unlike their first-order counterparts, which take only first-order derivatives into account, second-order optimizers will in addition incorporate the loss function's second-order features, or, in other words, the curvature information.

### 2.1 NEWTON'S METHOD AND ITS VARIANTS

Newton's method and its variants are perhaps the most typical second-order optimization skills. To be specific, in Newton's method, we approximate the loss function with its Taylor expansion up to order 2:

$$L(\boldsymbol{\theta} + \delta\boldsymbol{\theta}) \approx L(\boldsymbol{\theta}) + \nabla_{\boldsymbol{\theta}} L(\boldsymbol{\theta})^{\top} \delta\boldsymbol{\theta} + \frac{1}{2} \delta\boldsymbol{\theta}^{\top} \nabla_{\boldsymbol{\theta}}^2 L(\boldsymbol{\theta}) \delta\boldsymbol{\theta}$$

$$\triangleq L(\boldsymbol{\theta}) + \boldsymbol{g}(\boldsymbol{\theta})^{\top} \delta\boldsymbol{\theta} + \frac{1}{2} \delta\boldsymbol{\theta}^T \boldsymbol{H}(\boldsymbol{\theta}) \delta\boldsymbol{\theta}, \tag{1}$$

in which $\boldsymbol{g}(\boldsymbol{\theta})$ and $\boldsymbol{H}(\boldsymbol{\theta})$ are the gradient and Hessian matrix of $L$ at $\theta$ respectively. We then hope to move towards the direction in which the quadratic function on the right-hand side of Equation 1 is minimized. This condition implies that the standard Newton's method should proceed as

$$\boldsymbol{\theta}_{t+1} \leftarrow \boldsymbol{\theta}_t - \eta_t \boldsymbol{H}(\boldsymbol{\theta}_t)^{-1} \boldsymbol{g}(\boldsymbol{\theta}_t), \tag{2}$$

where $\eta_t$ is the learning rate for the $t$-th step. Here the Hessian $H$ is assumed to be invertible. The quantity $\boldsymbol{u} = \boldsymbol{H}^{-1}\boldsymbol{g}$ is usually called a Newton step.

Despite the theory that fewer steps are required for convergence, the original Newton's method has long been criticized for high computation and storage costs in calculating the Hessian matrix, inverting it, and other additional matrix manipulations (Nocedal & Wright, 1999; Boyd et al., 2004). To remedy this deficiency of speed, amazing increments of Newton's method have been proposed successively in recent centuries. Included are those reducing the cost by using an approximation of the Hessian matrix (Levenberg, 1944; Marquardt, 1963; Botev et al., 2017), and those quasi-Newton optimizers such as BFGS (Broyden, 1970; Fletcher, 1970; Goldfarb, 1970; Shanno, 1970) and its limited-memory version L-BFGS (Byrd et al., 1995) which approximate the inverse of Hessian matrix directly. Those aforementioned methods can partly alleviate the intractability of full Newton's method on deep neural networks.

### 2.2 NATURAL GRADIENT DESCENT AND K-FAC

In a common situation of supervised learning, a class of parametric conditional distribution $\{p(y \mid x, \boldsymbol{\theta}) \mid \boldsymbol{\theta} \in \Theta\}$ is assigned to the model to fit the underlying distribution $q(y \mid x)$ of the ob-

served data. Kullback-Leibler divergence (Kullback & Leibler, 1951) $L(\boldsymbol{\theta}) = \mathcal{D}_{KL}[q(x,y)\|p(x, y \mid \boldsymbol{\theta})]$ between the two joint distributions, or up to an additive constant, the negative log-likelihood, is usually selected as the loss function to be minimized. Natural gradient descent suggests an update direction satisfying

$$\boldsymbol{u}(\boldsymbol{\theta}) = \lim_{\varepsilon \to 0^+} \frac{1}{\varepsilon} \operatorname*{arg\,min}_{\mathcal{D}[p(x,y|\boldsymbol{\theta})\|p(x,y|\boldsymbol{\theta}+\delta\boldsymbol{\theta})] \leq \varepsilon^2} L(\boldsymbol{\theta}) \tag{3}$$

$$= -\kappa \boldsymbol{F}(\boldsymbol{\theta})^{-1} \nabla L(\boldsymbol{\theta}), \tag{4}$$

where $\kappa$ is a positive constant and

$$\boldsymbol{F}(\boldsymbol{\theta}) = \mathbb{E}_{p(x,y|\boldsymbol{\theta})}[\nabla \log p(x, y \mid \boldsymbol{\theta})^\top \nabla \log p(x, y \mid \boldsymbol{\theta})], \tag{5}$$

is known as the Fisher information matrix (FIM) of $p(x, y \mid \boldsymbol{\theta})$. Since the true FIM is generally not accessible, it is a common practice to evaluate the empirical FIM (Martens, 2020) instead, that is, to evaluate

$$\hat{\boldsymbol{F}}(\boldsymbol{\theta}) \triangleq \mathbb{E}_{\hat{q}(x,y)}[\nabla_{\boldsymbol{\theta}} \log p(x, y \mid \boldsymbol{\theta})^\top \nabla_{\boldsymbol{\theta}} \log p(x, y \mid \boldsymbol{\theta})], \tag{6}$$

in which $\hat{q}$ is the empirical distribution of the observed samples.

Interestingly, when current distribution $p(\cdot \mid \boldsymbol{\theta})$ and target distribution $q$ are close to each other, we have $\nabla_\theta^2 L(\boldsymbol{\theta}) \approx \boldsymbol{F}(\boldsymbol{\theta})$. This relation sheds light on the connection between Newton's method and natural gradient descent, in the sense that natural gradient descent can be conceived as a special Newton's method with approximate Hessian matrices. For this reason, we do not distinguish natural gradient descent from Newton's method in the rest of the paper and also call the update in equation 4 a *Newton step* or *Newton update*. Further discussion on the connections and distinctions among FIM, empirical FIM and Hessian matrix are provided in Kunstner et al. (2019).

Vanilla natural gradient descent suffers from the same inefficiency with full Newton's method. Compared to the numerous increments of Newton's method, few improvements for natural gradient descent have ever been devised, among which is the enlightening work of Kronecker-factored approximate curvature (K-FAC) (Martens & Grosse, 2015). K-FAC specializes in optimizing the weights only involved once in linear mapping, including the weights of fully-connected linear projection and convolutional operation (Grosse & Martens, 2016; Martens et al., 2018; Ba et al., 2016). The core idea of K-FAC is the approximate evaluation of empirical FIM $\hat{\boldsymbol{F}}(\boldsymbol{\theta})$ taking advantage of the properties of Kronecker products. To be specific, it approximates the expectation of the Kronecker product with the Kronecker product of expectation. In this way, the computation and storage burden is greatly relieved without simplifying the second-order structures too much. K-FAC has been proved a promising optimizer across various deep learning models and tasks (Martens & Grosse, 2015; Grosse & Martens, 2016; Martens et al., 2018; Wu et al., 2017; Osawa et al., 2019), and will be the major second-order optimizer we implement and analyze in this paper. More details about natural gradient descent and K-FAC can be found in appendix A.

## 3 PARAMETER-EFFICIENT TUNING ENABLES SECOND-ORDER TRAINING

In the foregoing section, we have introduced typical second-order optimizers together with incremental methods which ameliorate their efficiency in large-scale settings. In spite of that amelioration, second-order optimization still requires generally $N^2$ order of storage space and $N^3$ order of computing operations for every full parameter block with $N$ parameters, and is therefore intractable on extremely large models, especially those foundation language models based on deep transformers. For example, it is unlikely a possible mission to fine-tune a T5$_{\text{XL}}$ with either L-BFGS or K-FAC directly. The sad truth informs us that, beyond making second-order optimizers lighter and faster, more modifications should be made to the way we train. While hardly any further improvements could be made to the optimizers, cutting down the volume of the trainable parameters might be another way out. This straightforward thought coincides with a new paradigm in the spotlight – *parameter-efficient tuning* – including adapter, LoRA, and other useful methods, which we will explain in detail in the following passage.

### 3.1 BACKGROUND OF PARAMETER-EFFICIENT TUNING

Parameter-efficient tuning aims to adapt PTMs, especially large ones, via optimizing a tiny proportion of parameters. This intuition could also be found in previous works under other scenarios (Tajbakhsh

Table 1: Peak memory consumption of different models with first- and second-order optimization. #P denotes the number of total parameters, and #T denotes trainable parameters. All the numbers are measured on the RTE dataset with LoRA with a single NVIDIA A100 GPU, batch size = 16.

| Model | #P | #T | Optim. | Mem. |
|-------|-----|------|--------|----------|
| BERT | 336M | 0.75M | AdamW | 4.22 GB |
|       |      |       | K-FAC | 4.99 GB |
| RoBERTa | 355M | 0.75M | AdamW | 4.32 GB |
|         |      |       | K-FAC | 5.08 GB |
| T5$_{base}$ | 220M | 0.56M | AdamW | 2.05 GB |
|             |      |       | K-FAC | 2.49 GB |
| T5$_L$ | 770M | 1.50M | AdamW | 6.00 GB |
|        |      |       | K-FAC | 7.53 GB |
| T5$_{XL}$ | 3B | 3.75M | AdamW | 20.94 GB |
|           |    |       | K-FAC | 34.00 GB |

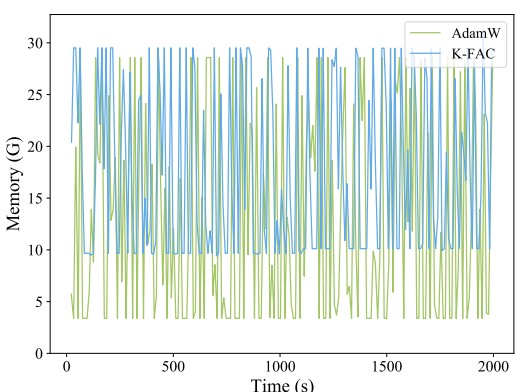

Figure 2: An example of GPU memory variation during training on the STS-B dataset with adapters. The used model is RoBERTa$_{large}$ and the batch size is 128. Both experiments are run on a single NVIDIA A100 GPU.

et al., 2016; Guo et al., 2020; 2019). The most practical advantage of such a paradigm is that we do not need to update all the parameters and produce separate fine-tuned instances for every downstream task. By training such lightweight parameters, we are able to flesh out the abstract ability of large-scale models to solve specific problems. Houlsby et al. (2019) injects small neural modules to each layer of the Transformer model and only optimizes the adapters in training. Subsequently, a series of variants of adapters have emerged (Pfeiffer et al., 2020; Sung et al., 2021; Mahabadi et al., 2021; Sung et al., 2022). Prefix and prompt tuning (Li & Liang, 2021; Lester et al., 2021) prepends tunable parameters to the input layers. LoRA (Hu et al., 2021) injects low-rank trainable decomposition matrices to the weights and is successfully applied to GPT-3 (Brown et al., 2020) with 175 billion parameters. Apart from introducing additional parameters, experiments show that optimizing a designated proportion of the inherent parameters produce a similar effect (Zhao et al., 2020; Zaken et al., 2021; Guo et al., 2021). This line of work implies that after massive pre-training, the adaptation of large-scale PTMs may be a "simple" process and is worth further exploring (He et al., 2022; Ding et al., 2022).

**Adapter.** The adapter (Houlsby et al., 2019) method inserts lightweight neural modules (adapter layer) to each layer of the Transformer model. Given the input hidden state $\boldsymbol{h}_{in} \in \mathbb{R}^d$, each adapter layer comprises a down-projection $\boldsymbol{D} \in \mathbb{R}^{d \times r}$, a non-linear activation function $\sigma(\cdot)$, and an up-projection $\boldsymbol{U} \in \mathbb{R}^{r \times d}$. There is also a residual connection from the input to the output of an adapter layer.

$$\boldsymbol{h}_{out} \leftarrow \sigma(\boldsymbol{h}_{in}\boldsymbol{D})\boldsymbol{U} + \boldsymbol{h}_{in}. \tag{7}$$

The position of adapter layers is optional. Houlsby et al. (2019) add them after the multi-head attention layer and the feed-forward layer of each Transformer block. There are also variants that add adapter modules to other positions like the LayerNorm layer (Pfeiffer et al., 2020). Depending on the choice of the size of the bottleneck dimension $r$, the tunable parameters of the adapter approach account for roughly 0.5%∼8% of the total number of parameters.

**LoRA.** The LoRA method (Hu et al., 2021) assumes that the change of each weight matrix is intrinsically low-rank, thus it injects tunable low-rank decomposition matrices $\boldsymbol{D} \in \mathbb{R}^{d \times r}, \boldsymbol{U} \in \mathbb{R}^{r \times d}$ to estimate the weight changes $\Delta \boldsymbol{W} = \boldsymbol{DU}$. Hence, the output hidden state is

$$\boldsymbol{h}_{out} \leftarrow \boldsymbol{h}_{in}(\boldsymbol{W} + \boldsymbol{D})\boldsymbol{U}, \tag{8}$$

where $\boldsymbol{W}$ is the initial weight matrix. Depending on the intrinsic rank $r$, the optimizable parameters of LoRA are normally less than 1% of the total parameters.

### 3.2 PARAMETER-EFFICIENT TUNING ENABLES SECOND-ORDER TRAINING

Having introduced the background in second-order optimization and parameter-efficient tuning and clarified our motivation for integrating them, we move on to verify through experiments that

parameter-efficient tuning can indeed enable second-order optimization on large-scale PTMs. As shown in Table 1, PTMs with different architectures and scales could be tractably adapted with a single NVIDIA A100 GPU. It is worth noting that the excess memory consumption resulting from second-order optimization is relatively low compared to the whole memory consumption. As the model scales, the absolute amount of parameters to be fine-tuned increases, and the memory footprint of the second-order optimization increases considerably. For example, in T5$_{XL}$, the second-order optimization takes up 13.06 GB more video memory than the first-order optimization. However, this is still acceptable in practice. We report an example of the GPU memory variation in Figure 2, which has a jagged shape over the training time. At low points, all the data is sent to the CPU for processing, leaving only the parameters of the model itself and the states in the optimizer in GPU. The gap between the first- and second-order optimization's memory usage at the low point reflects the volume difference in states of optimizers. We also conduct an analysis of time efficiency in Appendix C.3.

## 4 NEWTON-STEP CLIPPING STABILIZES SECOND-ORDER TRAINING

The combination of second-order optimization and parameter-efficient tuning is not enough for training smoothly (illustrated in Figure 4 in Section 5.2). We reveal both in theory and by experiments that one new auxiliary skill, which we call Newton-step clipping, has to be implemented to achieve satisfactory results. In this section, we first introduce a more traditional stabilizing strategy named gradient clipping. We will afterward clarify the divergence between gradient clipping and Newton-step clipping and justify our initiative of devising and applying this new approach.

### 4.1 GRADIENT CLIPPING

One crucial point for a smoother training experience is to manage your step-sizes (norms of update tensors) wisely. However, unlike the cases in toy tasks, it is of low efficiency to apply delicate step-size schedules, such as line search with Wolfe (Wolfe, 1969) and Armijo-Goldstein (Armijo, 1966) conditions, to large models like a deep-transformer-based PTM. Such low efficiency is partially ascribed to high computational cost, and, more substantially, to the nonconvex landscape that invalidates the theoretical benefit of those schedules. Mostly adopted in deep learning is to set up a fixed learning rate schedule for all layers at the beginning of a training process. But, this solution may not be flexible enough and may result in gradient explosion problem since the optimal step-sizes for different layers can typically be different.

Two strategies – adaptive gradient and gradient clipping – have been devised in previous works to tackle this issue. AdaGrad (Hinton et al., 2011), RMSProp (Duchi et al., 2012), Adam (Kingma & Ba, 2014), and other optimizers adopting the adaptive gradient strategy divide the first-order term by the square root of the second-order term to ensure that the magnitude of each layer's step-size automatically remains in a rather stable range highly irrelevant to the gradient norm. Compared to the adaptive gradient methods, gradient-clipping (Pascanu et al., 2013; Goodfellow et al., 2016) is a more straightforward approach to set off the influence of extremely rugged landscapes. In this approach, a gradient vector is clipped whenever its norm exceeds a fixed threshold, that is,

$$\boldsymbol{g}^{clp} \leftarrow \min\left\{1, \frac{\tau}{||\boldsymbol{g}||}\right\} \boldsymbol{g}. \tag{9}$$

This approach is designed mostly by intuition to conquer the high-curvature landscape in deep learning, especially indispensable for some NLP-specialized models (Pascanu et al., 2013). Apart from the stabilizing effect, more of its advantages are subsequently explored and proved, such as faster convergence (Zhang et al., 2019) and prevention of convergence to stationary points (Chen et al., 2020).

### 4.2 NEWTON-STEP CLIPPING

In fact, the term "gradient clipping" can be misleading. We claim that the underlying purpose of performing "gradient clipping" is to restrict the scale of the update tensor, NOT the gradient as is literally suggested. It is noteworthy that gradient clipping is usually applied to first-order methods, especially SGD and its variants. For these optimizers, the stepsize is proportional to the gradient norm, and hence clipping the gradient is equivalent to restricting the scales of updates. Nevertheless, for second-order optimizers, this rule may not hold. For instance, in natural gradient descent, the update is chosen to be

$$\boldsymbol{u} = \mathbb{E}[\boldsymbol{g}\boldsymbol{g}^\top]^{-1} \cdot \mathbb{E}[\boldsymbol{g}]. \tag{10}$$

---

**Algorithm 1:** Generalized procedures of pre-clipping and post-clipping

---

**Data:** Total iteration $T$, current iteration $t$, current learning rate $\eta_t$;

**while** $t \leq T$ **do**

  Compute and store gradient $\boldsymbol{g}_t$ and other required intermediate quantities $\boldsymbol{h}_t$;

  **if** *pre-clipping* **then**

    $\boldsymbol{g}_t \leftarrow \min\left\{1, \frac{\tau}{||\boldsymbol{g}_t||}\right\} \boldsymbol{g}_t$;

  Compute other required quantities $\boldsymbol{r}_t$;

  Transform the gradient to the final update by: $\boldsymbol{u}_t \leftarrow \mathcal{T}(\boldsymbol{g}_t, \boldsymbol{h}_t)$, where $\mathcal{T}$ is a given transform;

  **if** *post-clipping* **then**

    $\boldsymbol{u}_t \leftarrow \min\left\{1, \frac{\tau}{||\boldsymbol{u}_t||}\right\} \boldsymbol{u}_t$;

  Update the parameter $\boldsymbol{\theta} \leftarrow \boldsymbol{\theta} - \eta_t \boldsymbol{u}_t$.

---

Hence, we may expect that approximately $\|\boldsymbol{u}\| = \mathcal{O}(1/\|\mathbb{E}[\boldsymbol{g}]\|)$.

The above relation indicates that K-FAC can take an arbitrarily large step at a low curvature point, therefore, it is problematic to associate large gradient norm with large step size in natural gradient descent. A similar problem persists among general second-order optimizers due to the precondition of gradients by either Hessian matrices or FIMs. In addition, though traditional gradient clipping (that is applied before preconditioning) indeed shrinks the updates of a second-order optimizer to a certain extend, the preconditioned updates remain poorly controlled since the eigenvalues of Hessian matrices or FIMs can be arbitrarily small.

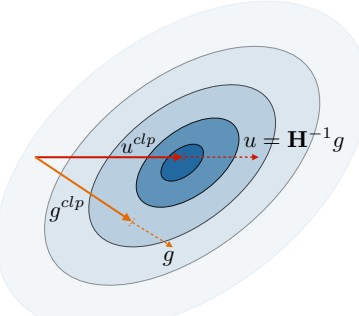

Figure 3: An illustration of the Newton-step clipping strategy.

For the above reasons, gradient clipping, though intuitively correct, may fails to improve the stability in second-order optimization. To address this issue, we propose Newton-step-clipping in which we clip the Newton update $\boldsymbol{u} = \boldsymbol{H}^{-1}\boldsymbol{g}$ (or $\boldsymbol{u} = \boldsymbol{F}^{-1}\boldsymbol{g}$) rather than the original gradient $\boldsymbol{g}$, that is,

$$\boldsymbol{u}^{clp} \leftarrow \min\left\{1, \frac{\tau}{||\boldsymbol{u}||}\right\} \boldsymbol{u}. \tag{11}$$

$\tau$, standing for the maximum update norm, is a hyperparameter. This approach has previously appeared as an exclusive skill for K-FAC training (Ba et al., 2016), but it would hopefully work for a wider range of second-order optimizers. We examine this approach on K-FAC, on which we discover that $\tau = 0.1$ generally works well.

### 4.3 GENERALIZATIONS: PRE-CLIPPING AND POST-CLIPPING

We can generalize our approach as follows. In gradient-based optimization, gradient plays a major role. However, the weight (or parameter) is seldom updated directly by its corresponding gradient; most optimizers will further transform the gradient into a final update tensor. Newton's method makes a case in point. It left-multiplies the gradient with the inverse of Hessian to yield a final Newton step. Then a divergence emerges in the question that when is the appropriate time for clipping. The vanilla gradient clipping strategy clips the gradients at the stage when the gradient backpropagation are completed. This sounds reasonable for SGD, feasible for Adam, but problematic for K-FAC. A different measure is not to clip anything until the final updates have been transformed from the original gradients, such as our Newton-step clipping. According to their time of clipping, we name the two aforementioned approaches pre-clipping and post-clipping by convenience. Procedures of pre-clipping and post-clipping are summarized in algorithm 1.

In a word, we have explained in the above passage that the choices of when to clip and what to clip can lead to entirely different training results. Although second-order optimizers favor later clipping upon the Newton step, for most of the rest of the optimizers that have ever been proposed, the choices between pre-clipping and post-clipping remain a mystery to be explored in the future.

## 5 EVALUATION

In this section, we evaluate and analyze our approach with corresponding baselines across widely used natural language understanding (NLU) tasks and different backbone PLMs.

### 5.1 EXPERIMENTAL SETTINGS

**Datasets.** The evaluated benchmarks in our experiments include SST-2 (Socher et al., 2013) for sentiment analysis, MRPC (Dolan & Brockett, 2005) for paragraph detection, CoLA (Warstadt et al., 2019) for inference acceptability, RTE (Dagan et al., 2005) for inference, QNLI (Rajpurkar et al., 2016) for inference, STS-B (Cer et al., 2017) for textual similarity, Choice of Plausible Alternatives (COPA) (Gordon et al., 2012) for commonsense causal reasoning, CommitmentBank (CB) (Marneffe et al., 2019) for inference, and Winograd Schema Challenge (WSC) (Levesque, 2011) for commonsense reasoning.

**Setup.** We adopt RoBERTa$_{large}$ (Liu et al., 2019) with 350 million parameters for QNLI, SST-2, RTE, COLA, MRPC, and STS-B; and T5-3b (Raffel et al., 2020) with 3 billion parameters for COPA, CB, RTE, and WSC. The project and second-order optimizers are implemented by PyTorch (Paszke et al., 2019), and the used models are loaded from the Huggingface Transformers (Wolf et al., 2019) library. We use NVIDIA Tesla A100 with 40GB memory for all experiments. Most datasets use accuracy as evaluation metrics, except for MRPC, CoLA, and STS-B. MRPC uses F1 scores. CoLA uses Matthew correlation coefficient (Matthews, 1975). Spearman correlation and Pearson correlation scores are reported on STS-B. More experimental details are reported in Appendix B.

### 5.2 EFFECT OF NEWTON-STEP CLIPPING

We first directly assess the effect of the Newton-step clipping method in the second-order training of deep transformers. When straightforwardly applying K-FAC to deep transformers without any clipping or with traditional pre-clipping, the training process is exceedingly unstable. Series of experimental results shown in Figure 4 verify that for large-scale second-order optimization, the strategy of Newton-step clipping significantly stabilizes the training process and outperforms its non-clipping counterpart, while pre-clipping cannot deliver similar effectiveness. The results indicate the indispensability of appropriate clipping when applying second-order optimization in the parameter-efficient paradigm, which is consistent with the discussion in Section 4.2.

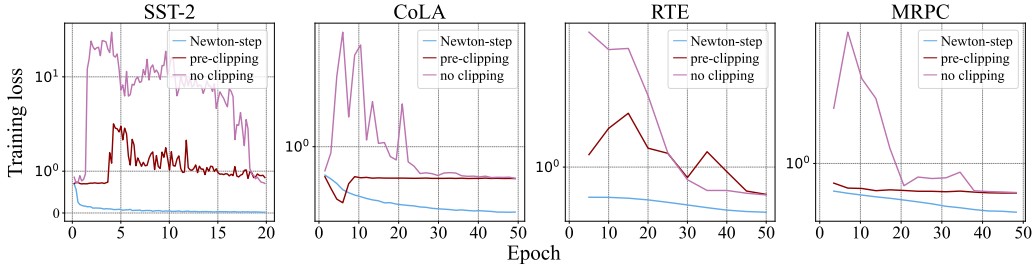

Figure 4: Training loss curve of K-FAC optimizer on RoBERTa$_{large}$ + Adapter with and without our clipping approaches. All settings and hyperparameters are the same except for clipping strategy.

### 5.3 EXPERIMENTAL RESULTS

Experimental results of natural language understanding are reported in Table 2. We mainly compare our approach with AdamW (Loshchilov & Hutter, 2017), which is broadly considered as the most powerful optimization method for deep transformers. Most of the parameter-efficient adaptations achieve on-par performances to full parameter fine-tuning. With both the adapter and LoRA approaches, second-order optimization with our Newton-step clipping considerably outperforms the AdamW counterparts. Specifically, in direct comparisons, with the adapter method, the average performance in six tasks of NewtwonClip outperforms AdamW by 1.06%. And with the LoRA method, the absolute improvement is 1.25%. We also observe that the performance gap is mainly reflected in RTE, COLA, and MRPC datasets, which are also generally considered to be the more difficult natural language understanding task than the other three tasks.

Table 2: Results on NLU tasks. † indicates the results are from Hu et al. (2021), and $a$, $b$, and $c$ indicate different amounts of trainable parameters. Blue and orange represent the best and second best performances of each column.

| Method | #Trainable params | QNLI | SST-2 | RTE | COLA | MRPC | STS-B | Avg. |
|---|---|---|---|---|---|---|---|---|
| RoBERTa$_{base}$ | 125M (100%) | 92.8 | 94.8 | 78.7 | 63.6 | 90.2 | 91.2 | 85.22 |
| RoBERTa$_{large}$ | 355M (100%) | 94.7 | 96.4 | 86.6 | 68.0 | 90.9 | 92.4 | 88.17 |
| *RoBERTa$_{large}$ + Adapter* | | | | | | | | |
| AdamW$^{\dagger a}$ | 6.0M (1.69%) | $94.7_{\pm 0.2}$ | $96.2_{\pm 0.3}$ | $83.4_{\pm 1.1}$ | $66.5_{\pm 4.4}$ | $88.7_{\pm 2.9}$ | $91.0_{\pm 1.7}$ | 86.75 |
| AdamW$^{\dagger b}$ | 3.0M (0.84%) | $94.8_{\pm 0.2}$ | $96.1_{\pm 0.3}$ | $83.8_{\pm 2.9}$ | $68.3_{\pm 1.0}$ | $90.2_{\pm 0.7}$ | $91.5_{\pm 1.5}$ | 87.45 |
| AdamW$^{\dagger c}$ | 0.8M (0.23%) | $94.7_{\pm 0.2}$ | $96.3_{\pm 0.5}$ | $72.9_{\pm 2.9}$ | $66.3_{\pm 2.0}$ | $87.7_{\pm 1.7}$ | $91.5_{\pm 0.5}$ | 84.90 |
| AdamW | 2.4M (0.67%) | $94.5_{\pm 0.3}$ | $95.6_{\pm 0.2}$ | $84.6_{\pm 2.2}$ | $64.9_{\pm 8.2}$ | $91.8_{\pm 1.2}$ | $91.9_{\pm 0.5}$ | 87.22 |
| NewtonClip | 2.4M (0.67%) | $94.3_{\pm 0.4}$ | $96.3_{\pm 0.4}$ | $86.8_{\pm 1.5}$ | $68.4_{\pm 1.3}$ | $92.3_{\pm 1.9}$ | $91.6_{\pm 0.3}$ | **88.28** |
| *RoBERTa$_{large}$ + LoRA* | | | | | | | | |
| AdamW$^{\dagger}$ | 0.8M (0.23%) | $94.8_{\pm 0.3}$ | $96.2_{\pm 0.5}$ | $85.2_{\pm 1.1}$ | $68.2_{\pm 1.9}$ | $90.2_{\pm 1.0}$ | $92.3_{\pm 0.5}$ | 87.82 |
| AdamW | 0.8M (0.23%) | $94.1_{\pm 0.6}$ | $95.4_{\pm 0.7}$ | $82.5_{\pm 0.9}$ | $69.1_{\pm 2.7}$ | $91.1_{\pm 0.2}$ | $91.6_{\pm 0.7}$ | 87.30 |
| NewtonClip | 0.8M (0.23%) | $94.4_{\pm 0.1}$ | $96.2_{\pm 0.3}$ | $85.7_{\pm 1.7}$ | $70.4_{\pm 1.8}$ | $92.7_{\pm 0.3}$ | $91.9_{\pm 0.3}$ | **88.55** |

As illustrated in Figure 5, equipped with Newton-step clipping, K-FAC demonstrates faster and more stable convergence than the first-order optimizer AdamW. Despite the effectiveness, the hyperparameter tuning of K-FAC is slightly more costly than that of AdamW since the second-order optimizer itself is more sensitive to the learning rate and also involves more hyperparameters, such as the scale of clipping. We carry out the ablation study of hyperparameters in Appendix C.1. We will investigate adaptive techniques for Newton-step clipping in future work to better deploy second-order optimization to PTM adaptations.

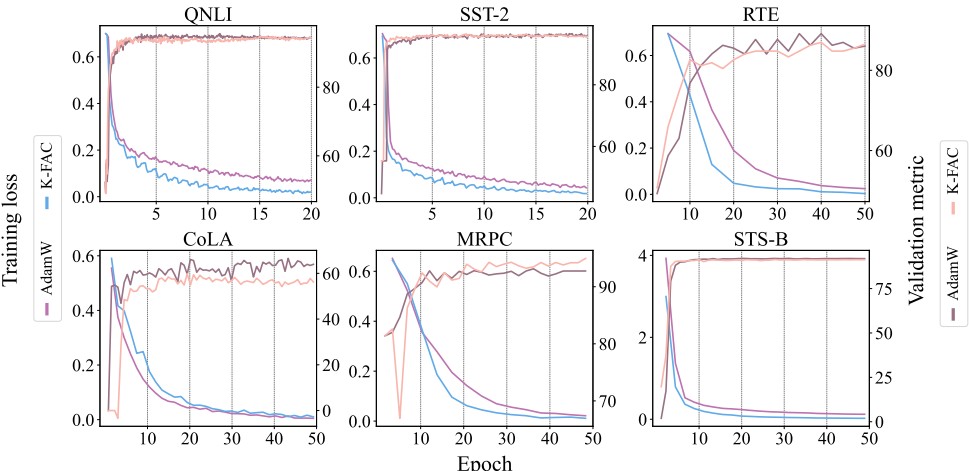

Figure 5: Training loss and validation metric curves of RoBERTa$_{large}$ + Adapter on NLU tasks with AdamW optimizer and K-FAC optimizer equipped with our Newton-step clipping strategy.

## 5.4 IMPACT OF THE NUMBER OF TRAINABLE PARAMETERS

To explore the impact of trainable parameters to our method, we train a RoBERTa$_{large}$ + LoRA with different LoRA intrinsic ranks (i.e., the bottleneck dimension $r$ of $\boldsymbol{D}$, $\boldsymbol{U}$ in equation 8) on the MRPC dataset. When linearly choosing $r$ in $\{8, 12, 16, 20, 24, 28, 32\}$, the amount of trainable parameters becomes $\{0.8M, 1.1M, 1.5M, 1.9M, 2.3M, 2.6M, 3.0M, 3, 4M\}$, respectively (we conduct 3 runs with different random seeds for each LoRA rank). However, as illustrated in Figure 6 and Figure 7, the convergence speed is observably inversely proportional to the number of trainable parameters, i.e., the smaller the number of trainable parameters, the faster the convergence speed. Meanwhile, we observe that the test performance does not change significantly as the number of trainable

parameters changes. For a PTM and a specific task, the adaptation process is "simple" that it can be accomplished with very few parameter optimizations, but it is difficult to make a leap in adaptation performance by changing the number of training parameters. In other words, the pre-training, model structure, and scale of the model itself seem to determine the upper limit of practical adaptations. The fact that using fewer parameters leads to faster convergence is also a testament to the effectiveness of our Newton-step clipping approach.

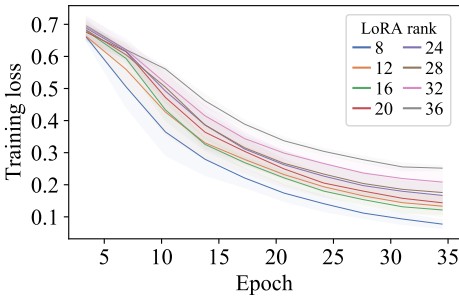

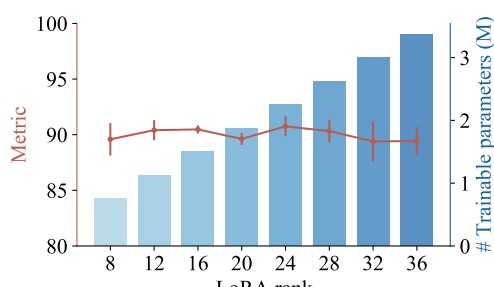

Figure 6: Training loss curve of models with different LoRA ranks. Experiments are conducted on MRPC with RoBERTa$_{large}$ and batch size is 128.

Figure 7: The change of test metrics (F1) and number of trainable parameters with different LoRA ranks. Experiments are conducted on MRPC with RoBERTa$_{large}$, batch size is 128.

### 5.5 SCALING TO 3 BILLION PARAMETERS

We scale our experimentation to T5$_{XL}$, a sequence-to-sequence model with 3 billion parameters. Evaluated datasets in this part are relatively small due to high training costs. As shown in Table 3, K-FAC with the proposed Newton-step clipping can achieve comparable performance with AdamW. For some datasets

Table 3: Results on COPA, CB, WiC, and WSC datasets of T5$_{XL}$ + Adapter.

| Method | COPA | CB | RTE | WSC | Avg. |
|---|---|---|---|---|---|
| AdamW | 82.00 | 89.29 | 79.86 | 67.31 | 79.62 |
| NewtonClip | 80.00 | 89.29 | 82.01 | 69.23 | **80.13** |

like RTE and WSC, our method even outperforms its first-order counterpart by a considerable margin. We also empirically find that larger models tend to favor larger maximum norms for Newton-step clipping due to the vast capacity. The success of K-FAC on T5$_{XL}$ further demonstrates its tractability under parameter-efficient tuning paradigm, and its potential in efficiently steering large pre-trained models.

## 6 DISCUSSION

Theoretical analysis and experimental results are presented in this paper, illustrating that the parameter-efficient paradigm can vivify second-order optimization on extremely large-scale PTMs with the assistance of the Newton-step clipping strategy. Although the application of second-order optimization on enormous PTMs is promising, the exploration is yet far from closed in the sense that pieces of dark clouds are still hanging over this topic. (1) To begin with, as we have observed in our experiments, second-order optimizers exhibit higher sensitivity to the choices of hyper-parameters compared to their first-order counterparts. While second-order optimizers tend to introduce more hyper-parameters, many of these newly-added hyper-parameters are more obscured in mathematical meaning and their experimental influence is more elusive. It remains unclear whether there are theories and methods to make the hyper-parameter tuning of second-order optimization no longer a sort of dark art. (2) Another uncertainty lies in the question that whether the design of architecture-specified optimizers is feasible. We notice in current work that both adapter and LoRA add to the original model fully-connected feed-forward branches which coincide with K-FAC's strength. But, for the other parameter-efficient methods like modifications of each head of attention output in Prefix Tuning, no similar conclusion has yet been drawn. (3) Moreover, it appears promising to study the combination of first-order or second-order optimization instead of sticking merely to one. Observed in our experiments is the rule of thumb that first-order optimizers, though with slower loss descent and lower test scores, enjoy better numerical and convergence stability. It is a natural deduction to run first-order steps as the warm-up period for second-order optimization. Seemingly trivial at first glance, the design of first and second-order compounds is bound to be an arduous journey of in-detail technical implementation.

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

# A  MORE ABOUT NATURAL GRADIENT DESCENT AND K-FAC

## A.1  CONNECTIONS BETWEEN NEWTON'S METHOD AND NATURAL GRADIENT DESCENT

The Hessian matrix of the Kullback-Leibler divergence (the loss function mentioned in section 2.2) reads

$$\nabla_{\boldsymbol{\theta}}^2 L(\boldsymbol{\theta}) = \nabla_{\boldsymbol{\theta}}^2 \mathcal{D}_{KL}[q(x,y)\|p(x,y\mid\boldsymbol{\theta})] \tag{12}$$

$$= -\mathbb{E}_{q(x,y)}[\nabla_{\theta}^2 \log p(x,y\mid\boldsymbol{\theta})] \tag{13}$$

$$= \mathbb{E}_{q(x,y)}[\nabla_{\theta}\log q(x,y)^{\top}\nabla_{\theta}\log p(x,y\mid\boldsymbol{\theta})] \tag{14}$$

In case that the current density $p(\cdot\mid\boldsymbol{\theta})$ is close to the target density $q$, we could see that $\nabla_{\boldsymbol{\theta}}^2 L \approx \boldsymbol{F}$.

## A.2  UPDATE RULES OF K-FAC

In accordance with the symbols adopted in Martens & Grosse (2015), a layer of linear mapping can be written as

$$\boldsymbol{s}_i = \boldsymbol{W}_i\overline{\boldsymbol{a}}_{i-1}. \tag{15}$$

Define $\boldsymbol{g}_i = \frac{\partial L}{\partial \boldsymbol{s_i}}$, then the $\boldsymbol{W}_i$-related sub-block of FIM can be expressed as

$$\begin{aligned}
\hat{\boldsymbol{F}}_{i,i} &= \mathbb{E}_{\hat{q}}[\mathrm{vec}(\nabla_{\boldsymbol{W}_i}L)\mathrm{vec}(\nabla_{\boldsymbol{W}_i}L)^{\top}] \\
&= \mathbb{E}_{\hat{q}}[(\overline{\boldsymbol{a}}_{i-1}\otimes\boldsymbol{g}_i)(\overline{\boldsymbol{a}}_{i-1}\otimes\boldsymbol{g}_i)^{\top}] \\
&= \mathbb{E}_{\hat{q}}[\overline{\boldsymbol{a}}_{i-1}\overline{\boldsymbol{a}}_{i-1}^{\top}\otimes\boldsymbol{g}_i\boldsymbol{g}_i^{\top}]
\end{aligned} \tag{16}$$

The core idea of K-FAC is to approximate the expectation of Kronecker product in equation 16 with the Kronecker product of expectation, that is,

$$\hat{\boldsymbol{F}}_{i,i} \approx \mathbb{E}_{\hat{q}}[\overline{\boldsymbol{a}}_{i-1}\overline{\boldsymbol{a}}_{i-1}^{\top}]\otimes\mathbb{E}_{\hat{q}}[\boldsymbol{g}_i\boldsymbol{g}_i^{\top}] \triangleq \boldsymbol{A}_{i-1}\otimes\boldsymbol{G}_i. \tag{17}$$

Furthermore, in light of the properties that $(\boldsymbol{B}\otimes\boldsymbol{C})^{-1} = \boldsymbol{B}^{-1}\otimes\boldsymbol{C}^{-1}$ and that $(\boldsymbol{B}\otimes\boldsymbol{C})\mathrm{vec}(\boldsymbol{V}) = \mathrm{vec}(\boldsymbol{C}\boldsymbol{V}\boldsymbol{B}^{\top})$, the natural gradient update in K-FAC can be formulated in tensors as

$$\boldsymbol{W}_i \leftarrow \boldsymbol{W}_i - \eta\cdot\boldsymbol{G}_i^{-1}(\nabla_{\boldsymbol{W}_i}L)\boldsymbol{A}_{i-1}^{-1}. \tag{18}$$

# B  ADDITIONAL EXPERIMENTAL DETAILS

**Hyper-parameters.** For experiments in Table 2, we perform a hyper-parameter grid search for both AdamW and K-FAC optimizers to select better-performing models. For AdamW, the search space for learning rate is $\{0.01, 0.001, 0.0001, 0.00003\}$ and for maximum gradient norm clipping scale is $\{0.1, 1.0, 10\}$. For K-FAC, in addition to the learning rate and the Newton-step clipping scale ($\tau$), we also set the damping factor to stabilize the matrix inverting operation (Levenberg, 1944; Marquardt, 1963). The search space for learning rate, Newton-step clipping scale, and damping factor are $\{0.01, 0.05, 0.1, 0.5\}$, $\{0.1, 1.0, 1.5, 2.0\}$, and $\{$1e-2, 1e-3, 1e-4, 1e-5, 1e-6$\}$, respectively. The search space of K-FAC is larger for AdamW for two reasons. (1) K-FAC has more hyper-parameters than AdamW, and we find that the damping factor has a considerable impact on the training. (2) Using AdamW to optimize deep transformers is extensively practiced in the community and our previous empirical studies, and we choose the set of reasonable learning rates and simply use commonly used values for other hyper-parameters like weight decay. However, there is little empirical evidence to provide a reasonable search space for K-FAC on large language models. The second-order update interval for K-FAC is set to 500 in our experiments, which could simultaneously yield promising performance and time efficiency. The batch size is 128 for Table 2 and Figure 5, and the training epochs and steps for each dataset is shown in Table 4.

The reported results in Table 2 and Table 3 use the following hyper-parameter settings in Table 5 and Table 6. For results in Figure 4, we adopt K-FAC as the optimizer and set the learning rate as 0.01, and set the damping factor as 1e-4 for both experiments. For all experiments, weight decay is set to 1e-4, and a linear scheduler with warm-up is adopted.

**Datasets.** All datasets are loaded with huggingface datasets (Lhoest et al., 2021). Since labels are not accessible for the test set, we manually split part of the data for testing. For small datasets (#samples < 10K), we randomly divide the validation set into halves, and use one half as the test set and one half as the validation set. For larger datasets (#samples > 10K), we randomly take 1K samples from the original training set as validation and the rest as the training set, keeping the original validation set as the test set. All models are trained on the training set and evaluated every 200 steps on the validation set, the checkpoint with the best performance on the validation set is kept for evaluation on the test set.

Table 4: Total training epochs and steps for each dataset for Table 2 and Figure 5.

| Dataset | Epochs | Total Steps |
|---------|--------|-------------|
| QNLI | 20 | 16220 |
| SST-2 | 30 | 15570 |
| RTE | 50 | 1000 |
| CoLA | 50 | 3350 |
| MRPC | 50 | 1450 |
| STS-B | 50 | 2250 |

When dealing with MRPC, RTE, and STS-B datasets, some works use the best model checkpoint on the MNLI dataset for initialization to boost the performance. In our empirical study, we do NOT use this strategy, and we use usual initializations for all the models. In our experimentation, the main parameters of PTMs are frozen, and we only optimize the trainable modules.

**GPU Memory.** The GPU memory statistics in Table 1 and Figure 2 come from PyTorch API and NVIDIA, respectively, so there might be a small inconsistency caused by NVIDIA's extra calculation of cache memory.

Table 5: The training hyper-parameters for K-FAC with the Newton-step clipping strategy.

| Model | Dataset | Method | Learning Rate | Max. Norm | Damping Factor |
|-------|---------|--------|---------------|-----------|----------------|
| RoBERTa$_{large}$ | QNLI | Adapter | 0.1 | 1.0 | 1e-4 |
| | | LoRA | 0.05 | 1.0 | 1e-3 |
| | SST-2 | Adapter | 0.1 | 1.0 | 1e-3 |
| | | LoRA | 0.1 | 1.0 | 1e-4 |
| | RTE | Adapter | 0.05 | 1.5 | 1e-3 |
| | | LoRA | 0.1 | 1.5 | 1e-5 |
| | CoLA | Adapter | 0.01 | 1.0 | 1e-2 |
| | | LoRA | 0.01 | 1.0 | 1e-2 |
| | MRPC | Adapter | 0.5 | 0.1 | 1e-2 |
| | | LoRA | 0.05 | 1.0 | 1e-3 |
| | STS-B | Adapter | 0.1 | 1.0 | 1e-2 |
| | | LoRA | 0.1 | 1.0 | 1e-6 |
| T5$_{XL}$ | COPA | Adapter | 0.1 | 10.0 | 1e-2 |
| | CB | Adapter | 0.1 | 10.0 | 1e-2 |
| | WSC | Adapter | 0.1 | 10.0 | 1e-3 |
| | RTE | Adapter | 0.1 | 10.0 | 1e-2 |

## C    ADDITIONAL EXPERIMENTAL RESULTS

This section reports additional experimental results to Section 5. We study the effect of hyper-parameters, time efficiency, and clipping occurrences in the training process. We find that the training will be more sensitive when dealing with small datasets. Hence, to take a closer look at the training procedure, we choose a larger dataset (STS-B and MRPC, respectively) and a small dataset (RTE) for analyses in Appendix C.1 and Appendix C.3,

### C.1    EFFECT OF HYPER-PARAMETERS

Hyper-parameters could considerably influence the performance of optimization. In our case, we find that the learning rate, maximum norm, and damping factor could simultaneously affect the training process. Hence, we conduct an ablation study to explore such impacts. When studying the effect of

Table 6: The training hyper-parameters for AdamW.

| Model | Dataset | Method | Learning Rate | Max. Norm |
|---|---|---|---|---|
| RoBERTa$_{large}$ | QNLI | Adapter
LoRA | 1e-4
1e-3 | 0.1
0.1 |
| | SST-2 | Adapter
LoRA | 1e-4
1e-3 | 0.1
0.1 |
| | RTE | Adapter
LoRA | 1e-4
1e-3 | 0.1
0.1 |
| | CoLA | Adapter
LoRA | 1e-3
1e-4 | 0.1
1.0 |
| | MRPC | Adapter
LoRA | 1e-4
1e-3 | 1.0
0.1 |
| | STS-B | Adapter
LoRA | 1e-4
1e-4 | 0.1
0.1 |
| T5$_{XL}$ | COPA | Adapter | 1e-4 | 0.1 |
| | CB | Adapter | 1e-4 | 0.1 |
| | WSC | Adapter | 1e-4 | 0.1 |
| | RTE | Adapter | 1e-4 | 0.1 |

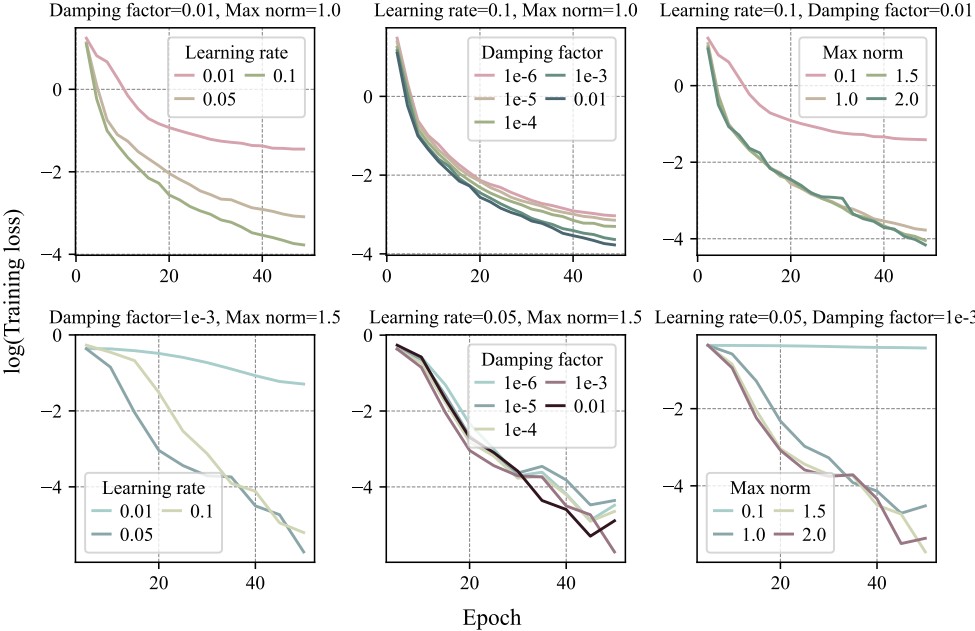

Figure 8: Impact of different hyper-parameters on test performance with STS-B dataset (upper) and RTE dataset (lower). Experiments are conducted with RoBERTa$_{large}$ + Adapter and K-FAC + NewtonClip, and is trained with batch size of 128 for 50 epochs.

one hyper-parameter, we fix the other two hyper-parameters. We run an ablation study on STS-B and RTE datasets and illustrate the training loss in Figure 8 and the corresponding performance on the test set in Table 7. It could be seen that with a small learning rate and a small maximum norm value, the model may fail to converge within a reasonable number of training steps. It indicates that, generally, the second-order optimizer is compatible with a relatively large update value, which is in line with its theoretical precision in loss landscape estimation. In terms of performance, it is relatively stable for STS-B, but observes a larger variation on the RTE dataset. It is probably because the scale of RTE is much smaller and thus the performance is more vulnerable to small perturbations in model parameters.

Table 7: Impact of different hyper-parameters on training loss with STS-B and RTE datasets. Experiments are conducted with RoBERTa$_\text{large}$ + Adapter and K-FAC + NewtonClip, and are trained with batch size of 128 for 50 epochs.

| Dataset | Learning Rate | Damping Factor | MaxNorm | Performance |
|---|---|---|---|---|
| STS-B | 0.01 | 1e-2 | 1.0 | 91.4 |
| | 0.05 | 1e-2 | 1.0 | 91.7 |
| | 0.1 | 1e-3 | 1.0 | 91.3 |
| | 0.1 | 1e-4 | 1.0 | 91.4 |
| | 0.1 | 1e-5 | 1.0 | 91.6 |
| | 0.1 | 1e-6 | 1.0 | 92.1 |
| | 0.1 | 1e-2 | 0.1 | 91.3 |
| | 0.1 | 1e-2 | 1.5 | 92.0 |
| | 0.1 | 1e-2 | 2.0 | 91.9 |
| | 0.1 | 1e-2 | 1.0 | 92.0 |
| RTE | 0.01 | 1e-3 | 1.5 | 71.2 |
| | 0.1 | 1e-3 | 1.5 | 83.5 |
| | 0.05 | 1e-2 | 1.5 | 80.6 |
| | 0.05 | 1e-4 | 1.5 | 77.0 |
| | 0.05 | 1e-5 | 1.5 | 79.9 |
| | 0.05 | 1e-6 | 1.5 | 81.3 |
| | 0.05 | 1e-3 | 0.1 | 57.6 |
| | 0.05 | 1e-3 | 1.0 | 83.5 |
| | 0.05 | 1e-3 | 2.0 | 77.7 |
| | 0.05 | 1e-3 | 1.5 | 87.1 |

## C.2 CLIPPING IN THE TRAINING PROCESS

We investigate how the proposed clipping technique affects the training procedure. Figure 10 shows how many times the Newton-step clipping actually occurs in a whole training procedure. It can be seen that clipping happens in almost every step in the early training stage, testifying to the importance of clipping. After considerable iterations of training, some steps could obtain a steady amount of updates without Newton-step clipping. The phenomenon persists through different datasets. It should also be noted that clipping is not equivalent to shrinking the learning rate. The functionality of clipping depends on the scale of the update norm so that the scaling effect can be adjusted dynamically according to the current update scale. Learning rate adjustment, however, is applied in a pre-defined manner that happens independent of the current update. Designing a suitable learning rate schedule for a second-order optimizer would require the knowledge of the change in update norm, and is thus extremely difficult given the rugged loss landscape of deep neural networks. Figure 9 further proves the point where simply decreasing the learning rate without clipping technique will not achieve satisfactory training, where we reduce the learning rate to K-FAC without Newton-step clipping and find that none of them could steadily converge.

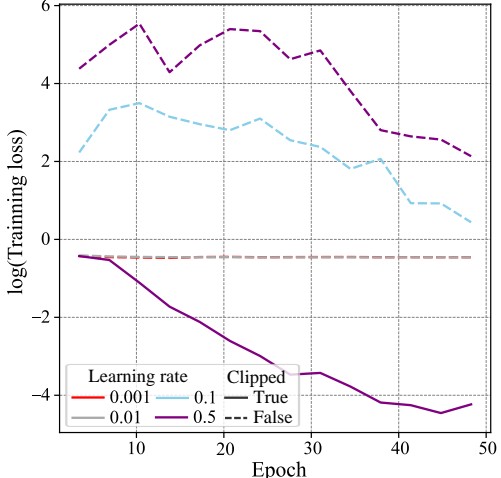

Figure 9: Comparison of training loss curve between small learning rate without post clipping and large learning rate with post clipping. Experiments are conducted on MRPC dataset with RoBERTa$_\text{large}$ + Adapter and K-FAC. Weight decay is 1e-4, and linear scheduler with warm-up is applied. The solid line denotes the training process with Newton-step clipping and dotted lines denote processes without Newton-step clipping across different learning rates.

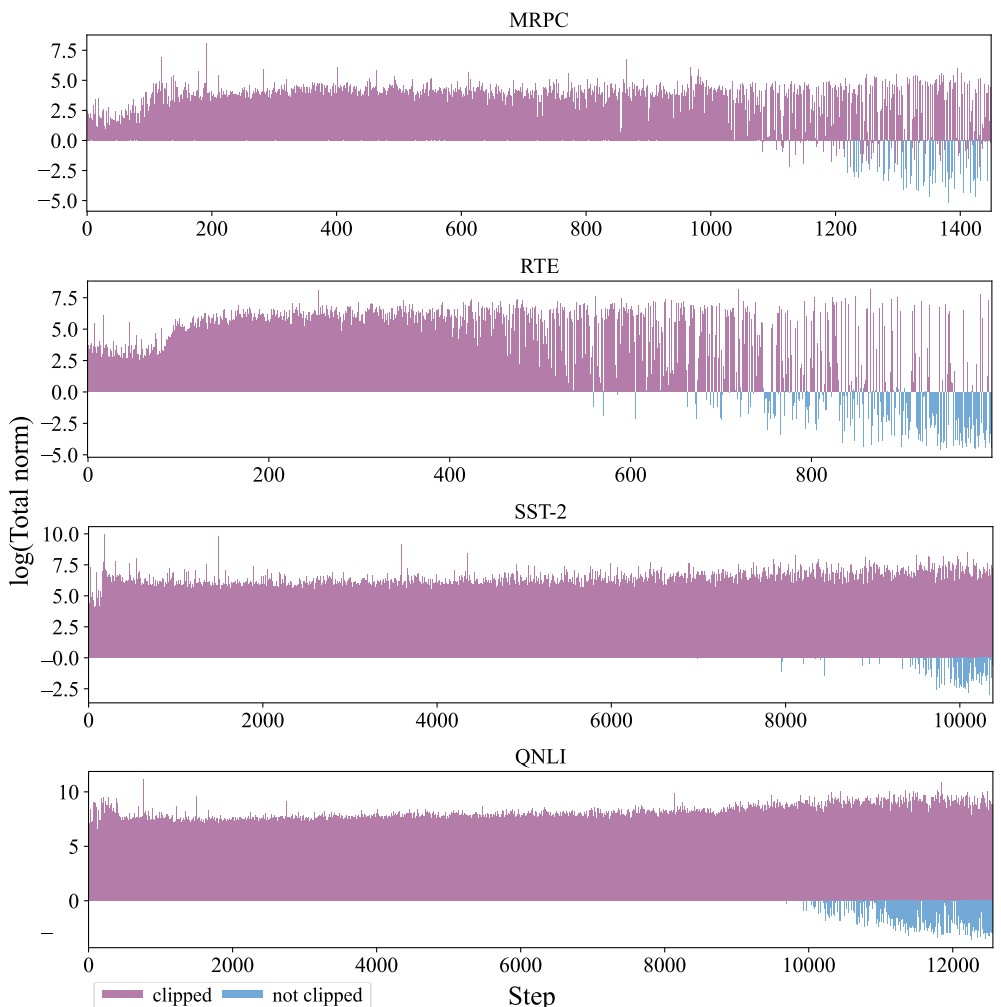

Figure 10: Total parameter norm and status of post clipping throughout the training process with RoBERTa$_{large}$ + Adapter and K-FAC + NewtonClip. For all datasets, models are trained with the optimum hyper-parameter setting.

### C.3 TIME EFFICIENCY

After analyzing the memory tractability of second-order optimization in Section 3.2, we find that second-order optimization is tractable for large language models under the parameter-efficient paradigm. In this section, we further conduct experiments for time efficiency. We conduct experiments on the MRPC and RTE datasets with a single NVIDIA A100 GPU and compute the average wall clock time over 100 epochs. For second-order optimization, the value of second-order update interval could particularly affect the training time. Thus we conduct different runs with intervals in {1, 50, 200, 500}. The results are reported in Table 8, from where we observe that K-FAC reaches comparable speed with AdamW when the update interval is set to 50 since there are not a lot of trainable parameters in the parameter-efficient paradigm. When the value is above 200, K-FAC outspeeds AdamW with satisfying performance (we did not fine-tune the hyper-parameters for each setting in this section). In the spirit of making second-order optimization as practical as possible on large models, we set it to 500 in all our other experiments.

Table 8: Wall clock time of AdamW and K-FAC + NewtonClip on the RTE dataset with RoBERTa$_{\text{large}}$. Values in the parentheses denote the relative time over AdamW.

| Dataset | Optimizer | Update Interval | Wall Clock Time | Performance |
|---|---|---|---|---|
| MRPC | AdamW | 1 | 31.3s (1.00×) | 91.3 |
| | K-FAC + NewtonClip | 1 | 56.3s (1.80×) | **92.7** |
| | K-FAC + NewtonClip | 50 | 33.6s (1.07×) | 90.7 |
| | K-FAC + NewtonClip | 200 | **27.9s (0.89×)** | 91.5 |
| | K-FAC + NewtonClip | 500 | 28.0s (0.89×) | 92.3 |
| RTE | AdamW | 1 | 18.4s (1.00×) | 84.6 |
| | K-FAC + NewtonClip | 1 | 31.4s (1.70×) | 81.3 |
| | K-FAC + NewtonClip | 50 | **16.3s (0.89×)** | 81.3 |
| | K-FAC + NewtonClip | 200 | 17.8s (0.96×) | 86.3 |
| | K-FAC + NewtonClip | 500 | 16.9s (0.92×) | **86.8** |

## C.4 PROLONGING TRAINING PROCESS

In this section, we take a closer look at the training process by prolonging the iterations. Specifically, we extend the training procedure to 500 epochs on the STS-B dataset, and the training loss and validation metric curves are shown in Figure 11. It can be seen that with both optimizers, the training loss keeps decreasing, while K-FAC observes an even smaller loss scale and converges faster, especially in the later training stage. At the final stage of training, both optimizers reach very small orders of magnitude in terms of training losses ($10^{-4}$ and $10^{-6}$ for AdamW and K-FAC, respectively). The validation metric curve of comparable performances also shows that there are no overfitting issues for both two optimizers, which could be regarded as an advantageous characteristic of parameter-efficient tuning.

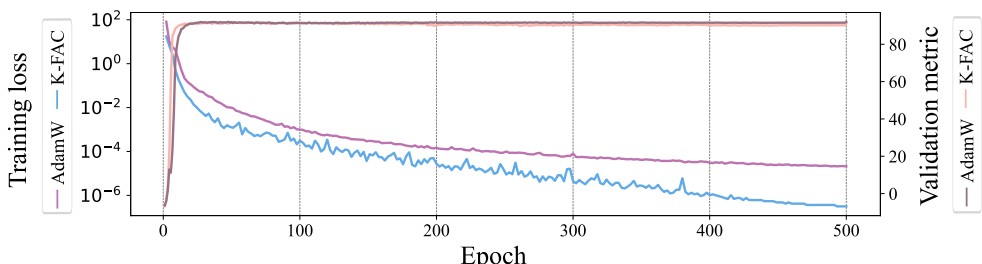

Figure 11: Training loss and validation metric curves on STS-B dataset with RoBERTa$_{\text{large}}$ over 500 training epochs. The hyper-parameter settings follow Table 5 and Table 6.

