# OpenReview forum: "Applying Second Order Optimization to Deep Transformers with Parameter-Efficient Tuning"
_ICLR.cc/2023/Conference — Submitted to ICLR 2023_

### Official Review · Reviewer_fdsh · 2022-10-23

**Confidence:** 4
**Correctness:** 2
**Technical Novelty And Significance:** 1
**Empirical Novelty And Significance:** 2
**Recommendation:** 3

**Clarity, Quality, Novelty And Reproducibility:**

As I mentioned in “Weaknesses,” some experiment details are unclear. This raises questions about the quality of the comparison and makes it difficult to reproduce the results. The technical novelty of this work seems limited.


**Strength And Weaknesses:**

Strengths
- Second-order optimization methods, which have computational and memory overhead, are suitable for parameter-efficient tuning. The numbers of the tunable parameters and the memory overhead of a second-order optimization method (I assume it is K-FAC) in fine-tuning pre-trained Transformer models presented in this work (Table1) help highlight this point.

Weaknesses
- The technical significance is limited.
    - With parameter-efficient tuning, the training time could already be short (this is what parameter-efficient tuning is for), even with first-order optimization. The low computational and memory cost of second-order optimization is attractive, but (as I also describe below “the validity of the comparison”) it is questionable whether the benefit of the second-order information is worth the cost in parameter-efficient tuning.
    - The proposed “Newton-step clipping” merely shifts the timing (from pre- to post-preconditioning) at which gradient clipping is applied. Post-preconditioning clipping of K-FAC gradient based on the KL-divergence of the model’s predictive distribution before and after the update (rather than the Euclidean norm of the update as in the gradient clipping) has already been proposed (https://jimmylba.github.io/papers/nsync.pdf). How one should determine the timing (pre- or post-) and "distance" (e.g., KL-divergence, Euclidean norm) is an interesting question, but that is not studied in this work.

- The validity of the comparison is questionable.
    - It is not clear whether the number of epochs shown in Figure 5, which depends on the task, is reasonable since the timing at which training stops (e.g., when training loss reaches a threshold, a fixed number of epochs) is not stated. In this case, there appears to be room for AdamW to achieve better performance when the number of epochs is further increased, and its difference from K-FAC could be more negligible.
    - Furthermore, there is no mention of how the model is selected for evaluation (e.g., using a validation set) or how the training set, validation set, and test set are partitioned, so it is doubtful that hyperparameters are tuned on the test set. (Just because training loss converges faster, as shown in Figure 5, does not necessarily mean that the model generalizes well.)
    - It is unclear whether K-FAC is "faster" than AdamW since there is no comparison regarding the wall-clock time for training. K-FAC is not necessarily faster than AdamW in wall-clock time (= number of steps x time per step) because of the per-step computational overhead compared to first-order optimization methods.
    - Finally, optimization methods should be compared not only for best results but also for their sensitivity to the choice of hyperparameters. It is difficult to consider an optimization method practical for other tasks if it is too sensitive.

Other comments
- Table1: what are the “first-order optimization” and “second-order optimization”? A reader can guess they are SGD (or AdamW) and K-FAC, but it is unclear.
- Eq (19) 2nd line: I believe it should be (a \times g)(a \times g)^T. The shape of each matrix and vector should be described to avoid confusion.
- p6 the last paragraph, “derivative-based optimization”: I believe this should be “gradient-based optimization.”

**Summary Of The Paper:**

This study applies second-order optimization (K-FAC) in parameter-efficient tuning (with adapters and LoRA) of Transformer models. Measurements show that the memory cost of second-order optimization can be relatively small due to the small number of trainable parameters, suggesting a second-order method is more suitable in this setting. This study proposes "Newton-step clipping," in which gradient clipping is applied after preconditioning, and shows that the training with K-FAC is stable. In the parameter-efficient tuning of the RoBERTa large model, K-FAC achieves lower training loss than AdamW for a given number of epochs.

**Summary Of The Review:**

The use of second-order information in parameter-efficient tuning is more reasonable than in full-parameter tuning, but its practical benefit is not clearly demonstrated in this study. The novelty of the proposed clipping procedure is limited, and its justification is more of an intuitive interpretation than a "theoretical" one. Based on the above, I do not believe that this study at this time is ready for publication.

---

> ### Author Response · Authors · 2022-11-11
> **Response to Reviewer fdsh (2/2)**
>
> > How one should determine the timing (pre- or post-) and "distance" (e.g., KL-divergence, Euclidean norm) is an interesting question, but that is not studied in this work.
> - Thanks for this thought-provoking question.
> - For the clipping time. We prove empirically that the choice of clipping time can make a great difference, and that for second-order optimization such as K-FAC, post-clipping can yield better performance than pre-clipping (demonstrated in Figure 4).
> - For the clipping distance. Our implementation is based on L2 norm as opposed to KL-divergence used in previous work. We only show with experiments that clipping the L2 norm leads to delightful results in our setting. We admit that, in our current work, the "choice of distance to clip" remains underexplored in more general cases, which calls for more theoretical and experimental investigations.
>
> > Furthermore, there is no mention of how the model is selected for evaluation (e.g., using a validation set) or how the training set, validation set, and test set are partitioned, so it is doubtful that hyperparameters are tuned on the test set. (Just because training loss converges faster, as shown in Figure 5, does not necessarily mean that the model generalizes well.
> - Thanks for your constructive suggestion, we have added these details in the new revision. We select our hyperparameter based on validation results as opposed to testing performance. No test data is involved in the training process. Table 2 reports the results in test datasets, which indicates the generalization power of our approach.
>
> > It is unclear whether K-FAC is "faster" than AdamW since there is no comparison regarding the wall-clock time for training. K-FAC is not necessarily faster than AdamW in wall-clock time (= number of steps x time per step) because of the per-step computational overhead compared to first-order optimization methods.
> - Thanks, we have supplemented a study of time efficiency in Appendix C.3 (Table 8). It is worth noting that second-order update intervals could considerably affect the training time, and we use interval = 500 in all our experiments, which turns out to be faster than AdamW.
>
> > Ablation study of the sensitivity of hyper-parameters.
> - Thanks, we agree that an ablation study is important, and we have supplemented it in Appendix C.1. For a relatively larger dataset (STS-B), the effect of hyper-parameters is not significant, while the model becomes more sensitive when dealing with a smaller dataset (RTE).
>
>
> > Table1: what are the “first-order optimization” and “second-order optimization”? A reader can guess they are SGD (or AdamW) and K-FAC, but it is unclear.
> - Fixed. We now clearly state that the first-order optimization is AdamW, and the second-order optimization is K-FAC.
>
> > Eq (19) 2nd line: I believe it should be (a \times g)(a \times g)^T. The shape of each matrix and vector should be described to avoid confusion.
> - You are right. Thanks for your suggestion.
>
> > P6 the last paragraph, “derivative-based optimization”: I believe this should be “gradient-based optimization.”
> - Fixed. Thanks for your suggestion.
>
>
> [1]. Delta tuning: A comprehensive study of parameter efficient methods for pre-trained language models.
>
> [2] Revisiting Parameter-Efficient Tuning: Are We Really There Yet?

---

> > ### Comment · Reviewer_fdsh · 2022-11-16
> > **Thank you for reply**
> >
> >
> > >  We deem it not quite meaningful to compare the limiting results in which the number of epochs is extensively increased.
> >
> > To be clear, I am not suggesting limiting to only extensive epoch number results.  I am suggesting comparing optimizers with various target epoch numbers (i.e., with various speeds of learning rate decay). For example, K-FAC might achieve a better accuracy than AdamW in X epochs, but AdamW might reach an even better accuracy in 2*X epochs, so there might be a time-accuracy trade-off (this is the case in Figure 11, where K-FAC achieves a better Validation metric until ~20 epochs, but AdamW overcomes it after). Therefore, when to stop training matters, and comparing only a specific number of epochs may lead to conclusions biased toward one method. Also, for this reason, setting a different number of epochs for each task in Figure 5 gives me an unfair impression.
> >
> > > (my comment) The proposed “Newton-step clipping” merely shifts the timing (from pre- to post-preconditioning) at which gradient clipping is applied.
> >
> > I would like to mention that the name “Newton-step clipping” is very confusing because it has little to do with Newton method and is not a good description of what it is. It is just a post-preconditioning gradient clipping, and I don’t think it is worth naming (highlighting the difference between pre- and post-preconditioning gradient clipping and its impact on the training is indeed important, though.)
> >
> >
> > I regret to point this out, but I think Table 8 (wall clock time vs. prediction performance comparison for AdamW and K-FAC) illustrates my concern about the motivation for this study very well. Due to the nature of parameter-efficient training, training time is extremely short, even with AdamW (31.3s and 18.4s). In such a small setting, one may find it is more effective to spend time on tuning hyperparameters of AdamW rather than introducing a second-order optimizer, which requires extra hyperparameters (matrix update interval, damping factor). Therefore, I am not convinced that a second-order optimizer is worth paying the implementation/tuning costs in parameter-efficient tuning, and I would like to keep my score.

---

> > > ### Author Response · Authors · 2022-11-16
> > > **Further Response to Reviewer fdsh**
> > >
> > > - Sorry for the confusion. To clarify, we set the numbers of training epochs purely based on previous empirical practice in the community, and our only goal is to make the models fully converge. For the GLUE benchmark, the common number of epochs ranges from 10 to 30 epochs, which could well ensure the convergence of deep transformers. The only reason that we train the model on SST-2 and QNLI for 20 epochs is that they are much larger than other datasets.  For example, according to the LoRA paper, the numbers of epochs for QNLI, SST-2, RTE, COLA, MRPC, and STS-B are 10, 10, 20, 20, 20, and 30, respectively. The six datasets can be regarded as separate tasks, and we believe our setting is reasonable. Although AdamW outperforms K-FAC on the STS-B validation data (we didn't hide it and already reported the test performance in Table 2), the gap is marginal since the performance of STS-B is relatively steady for the model with sufficient epochs of training. We never stop the model before enough training in our study. Hence, the empirical results under these numbers of epochs are reliable. Yes, when to stop the training matters, but it does not impactfully apply to our case since the training is sufficient.
> > > - We report Table 8 only to relatively compare the wall-clock time of different optimization methods. In fact, RTE and MRPC are extremely small NLP datasets (for example, the file size of RTE training data is 846KB with 2000+ instances), that's why the average time for an epoch is short. This is not because of parameter-efficient tuning, actually, the convergence time of parameter-efficient tuning is consistently slower than fine-tuning.
> > > - With the exact same pre-trained model and settings, boosting an average point on the GLUE benchmark just by merely changing the training strategy can be considered non-trivial since the model capacity already determines the lower bound of performance. And we must clarify that, our goal is not merely to find a "faster and better" optimization strategy for parameter-efficient tuning but to let the strength of second-order optimization extend to a more practical scenario. We will never know to what extent the finely derived superior properties of second-order optimization will work in modern deep transformers when we only evaluate it on CIFAR and ImageNet with CNN modules. New breakthroughs will be hard to come by when we experiment in a comfort zone due to computational complexity.
> > > - Our current work, as we have recapped in the general response, aims at realizing the tractability of second-order optimization on large pre-trained models via parameter-efficient tuning. It validates the possibility of designing, examining, and analyzing second-order optimizers on billion-scaled deep transformers, as opposed to relatively toy models which are more prevailing in the community.
> > > - We admit that the name "Newton-step clipping" can be misleading sometimes. Your suggestion of "post-(preconditioning)-clipping" can better describe its nature.
> > > - Sincerely, we thank you for your prompt reply. Your comment also circles some of our confusion and worry. Hope the above discussion can illustrate our motivation to you in a clearer manner.

---

> ### Author Response · Authors · 2022-11-11
> **Response to Reviewer fdsh (1/2)**
>
> Thanks for your constructive comments.
> > With parameter-efficient tuning, the training time could already be short (this is what parameter-efficient tuning is for), even with first-order optimization. The low computational and memory cost of second-order optimization is attractive, but (as I also describe below “the validity of the comparison”) it is questionable whether the benefit of the second-order information is worth the cost in parameter-efficient tuning.
>
> > It is not clear whether the number of epochs shown in Figure 5, which depends on the task, is reasonable since the timing at which training stops (e.g., when training loss reaches a threshold, a fixed number of epochs) is not stated. In this case, there appears to be room for AdamW to achieve better performance when the number of epochs is further increased, and its difference from K-FAC could be more negligible.
> - Thanks for this crucial comment. We deem it not quite meaningful to compare the limiting results in which the number of epochs is extensively increased. Previous works have shown a counter-intuitive rule that parameter-efficient tuning methods are generally slower than fine-tuning (even in wall-clock time), despite the fact that they only optimize trace amounts of parameters [1,2]. Thus, when devising optimization skills for parameter-efficient tuning, how to speed up convergence is more of an issue.  Plus, our current total number of epochs is relatively large compared to commonly adopted settings. And, within these epochs, our method can already attain better test performance.
> - The most important value of parameter-efficient tuning is that we only need to train and store lightweight modules rather than maintain a large model for each downstream task respectively (this paradigm could also save computational memory). For example, if we would like to adapt a GPT-3 model with 175B parameters to 100 downstream tasks, it is prohibitive to fine-tune 100 times and save 100 instances of GPT-3.
> - Nevertheless, we agree that AdamW is acceptable in training time. We would like to clarify that applying second-order optimization to large models is not merely about "getting a faster optimizer". On the model side, using second-order optimization may inspire better training methods and related analyses of giant language models. On the optimization side, we can evaluate the characteristics and practicability of previous 2nd-order optimizers under broader scenarios.
> - The training details are supplemented in Appendix B. We use fixed numbers of epochs that could ensure convergence in our experiments.
> - However, we agree that we can take a closer look by increasing the number of epochs, and we add this study in Appendix C.4. After training of 500 epochs, we surprisingly find that the loss is still decreasing under K-FAC + Newton-step clipping without loss of generalization performance. And we observe that K-FAC could yield a smaller loss scale ($10^{-6}$) than AdamW ($10^{-4}$), and both methods do not suffer from over-fitting.

---

### Official Review · Reviewer_xAvz · 2022-10-25

**Confidence:** 4
**Correctness:** 4
**Technical Novelty And Significance:** 2
**Empirical Novelty And Significance:** 2
**Recommendation:** 3

**Clarity, Quality, Novelty And Reproducibility:**

The paper is very clearly written.

My main concern is around novelty. The key ideas of fine-tuning with a small number of parameters, and clipping gradients are well studied in the literature. So the two new ideas here are --- using K-FAC instead of AdamW, and clipping gradients before and after preconditioning.
Neither are sufficiently novel to constitute an ICLR paper.

**Strength And Weaknesses:**

The main ideas in the paper are very easy to understand and well explained. The experimental section is good, with some caveats.

K-FAC is a carefully designed algorithm with extensive theoretical justification. How does NewtonClip affect this theory? Does the convergence rate change? Does the bias introduced by NewtonClip mess the independence assumptions?

Given that second order methods are more expensive computationally, the authors should include a comparison of the runtime and memory cost of AdamW with NewtonClip. Also, please state the results of training with K-FAC only (without clipping).

The statement "In spite of that amelioration, second-order optimization still requires at least N^2 ∼ N^3 order of storage space and computing operations" is not really correct for K-FAC --- in fact the storage is at most 2mN, where m is the largest dimension of any layer, so this is much less than N^2. Similarly, compute operations are at most m^2 N. Still large, but not N^3.

Page 7: NewtwonClip --> NewtonClip

**Summary Of The Paper:**

The paper makes two observations regarding training large language models:
1. During fine-tuning large transformers, several authors have previously shown that it is necessary to train a small number of parameters. This paper proposes using K-FAC to fine tune such models. The authors say that second order methods are very expensive for full training, but since fine-tuning involves few parameters, K-FAC is tractable, and achieves better results compared to first order methods.
2. Gradient clipping is a common method to prevent training blowups due to large gradients. The authors show that for K-FAC, this is not sufficient due to preconditioning, instead clipping should be done both before and after preconditioning.

The paper contains several experiments showing that a network fine-tuned with K-FAC achieves slightly better results than with AdamW.

**Summary Of The Review:**

The paper is currently not strong enough for ICLR. It can be strengthened by addressing some of the theoretical questions above, and slight enhancements to the experimental section.

---

> ### Author Response · Authors · 2022-11-11
> **Response to Reviewer xAvz**
>
> We sincerely thank you for your constructive comments.
> > Given that second-order methods are more expensive computationally, the authors should include a comparison of the runtime and memory cost of AdamW with NewtonClip.
> - The results of memory cost are reported in Table 1 and Figure 2. In fact, in Section 3, our goal is to validate the tractability of applying K-FAC on large language models with parameter-efficient tuning to ensure the follow-up experiments are runnable. More specifically, second-order optimization cannot be applied to large-scale models with full parameter fine-tuning due to computational consumption, we propose that instead of optimizing the complexity of second-order optimization itself, the latest parameter-efficient tuning results allow us to directly and drastically reduce the parameters to be updated, thus making the application of second-order optimization possible, and we verify the tractability in Section 3.
> - In this revision, we also supplement the analysis of time efficiency in Appendix C.3, which shows that in our experimental setting (second-order update interval = 500), the training is even slightly faster than AdamW without the loss of generalization.
> Please state the results of training with K-FAC only (without clipping).
> - The results with and without clipping on K-FAC are shown in Figure 4. It may not be easy to notice because we put them in Section 4 but not in Section 5. Now we have moved the results to Section 5.1, and they are still in Figure 4. We also supplement the results of K-FAC + pre-clipping in Figure 4 in the current revision. Also, an analytic experiment in Appendix C.2 (Figure 9) could also demonstrate its effectiveness.
>
> > The statement "In spite of that amelioration, second-order optimization still requires at least N^2 ∼ N^3 order of storage space and computing operations" is not really correct for K-FAC --- in fact the storage is at most 2mN, where m is the largest dimension of any layer, so this is much less than N^2. Similarly, compute operations are at most m^2 N. Still large, but not N^3.
> - Sorry for the confusion. In the aforementioned paragraph, we intend to state that for a full parameter block (usually one DNN layer) with $N$ parameters, the storage space and computing operations for most second-order optimizers (including K-FAC) will be in $N^2\sim N^3$ order. On the contrary, the conclusion of $2mN$ storage and $m^2N$ computation is based on the whole network. The reduction from $N^2$ to $2mN$ (and $N^3$ to $m^2N$) is not the contribution of K-FAC, but that of block-diagonal approximation -- which is also the default choice for a wide range of large-scale second-order optimizers. Following your suggestion, we have re-clarified in the new version that our statement is based on every single parameter block.
>
> > K-FAC is a carefully designed algorithm with extensive theoretical justification. How does NewtonClip affect this theory? Does the convergence rate change?
> - We show sincere respect towards the delicate design of K-FAC and other second-order optimization methods for large-scale training.  We provide only mostly experimental analysis in our application-oriented work, since we find it an arduous journey to devise theoretical justification under practical assumptions that actually hold for deep transformers. Do you consider it a good choice to start with stronger conditions?
>
> > Does the bias introduced by NewtonClip messes the independence assumptions?
> - NewtonClip will not affect the independence between $a$ and $g$ (please refer to the notation in Appendix A2 or the K-FAC paper). The reason is that the independence assumption is only related to the approximation of the Fisher matrix, while NewtonClip clips the final update after it has been preconditioned by the Fisher matrix.

---

### Official Review · Reviewer_Cqu9 · 2022-10-26

**Confidence:** 3
**Correctness:** 3
**Technical Novelty And Significance:** 2
**Empirical Novelty And Significance:** 2
**Recommendation:** 5

**Clarity, Quality, Novelty And Reproducibility:**

Presentation issues:
- The paper seems to be written quite informally, for instance "The mere combination of second-order optimization and parameter-efficient tuning is still far from training smoothly" or "clips the gradients as soon as the gradients are figured out". The paper's presentation can improve quite a bit if the authors would make their writing precise.
- Section 5.2: typo "NewtwonClip"

Reproducibility:
- I was unable to find # of training steps for the experiments in the appendix. I think it is useful to report it.
- The authors mention that "The search space for learning rate, Newton-step clipping scale, and damping factor are 1e − 2 ∼ 0.5, 0.1 ∼ 2.0 and 1e − 2 ∼ 1e − 6, respectively". To me, this is not specific enough. Please report either exact discrete points which were tested using a grid search or mention more details about an automatic tuner.


**Strength And Weaknesses:**

Strengths
- The general idea of leveraging expensive (but more effective) optimizers in the case of finetuning small number of parameters seems novel and practically useful.

Weaknesses
- It is unclear if tuning the learning rate more heavily may be able to subsume post-clipping, it would be interesting plot how many iterations clipping is actually applied. If it is applied in all iterations, then it may be corrected by just tuning the learning rate more thoroughly.
- It would also be interesting to apply clipping to the baseline AdamW method. The authors discuss a hand-wavy rationale for not doing it, but it does not seem satisfactory and I would be interested in well-tuned empirical results on clipping AdamW update.
- Another interesting experiment would be to run adamw longer and see if it can recover performance with Newton steps.

**Summary Of The Paper:**

The paper exploits a trend of low-rank finetuning or finetuning tiny number of parameters by applying a second order optimizer instead of typically used AdamW.

**Summary Of The Review:**

While the paper proposes an interesting idea of leveraging potentially more effective optimizers in the case of finetuning small set of parameters, the paper's contribution of clipping Newton step poses several unresolved questions and not quite fully supported through evidence.

---

> ### Author Response · Authors · 2022-11-11
> **Response to Reviewer Cqu9**
>
> We sincerely thank you for your  valuable comments.
> > It is unclear if tuning the learning rate more heavily may be able to subsume post-clipping, it would be interesting to plot how many iterations of clipping are actually applied. If it is applied in all iterations, then it may be corrected by just tuning the learning rate more thoroughly.
> - Thanks, it is a very interesting suggestion. Turning down the learning rate does not equal clipping strategies. Clipping is actually a simple yet flexible learning rate schedule depending on the norms of updates. It adopts bounded stepsizes at the early stage of training to avoid instability, without being too conservative to use infinitesimal lr which may cause poor convergence at the closing stage. Theoretically but only theoretically, the performance of clipping can be achieved by non-clipping under a delicately designed lr schedule with consideration to the varying norms. This is not practical in the least, since we generally do not know explicitly how the magnitude of gradient/update changes along the training trajectory of each task. We also prove through experiments that the performance of post-clipping cannot be recovered by tuning the learning rate only.
> - We agree that the analysis will be interesting. We plot Figure 10 in  Appendix C.2 to show that post-clipping is indeed performed for most of early steps in the training process. We also show that merely shrinking the learning rate of K-FAC cannot achieve the same performance in Figure 9 of Appendix C.2.
>
> > It would also be interesting to apply clipping to the baseline AdamW method. The authors discuss a hand-wavy rationale for not doing it, but it does not seem satisfactory, and I would be interested in well-tuned empirical results on clipping AdamW update.
> -  It is indeed a thought-provoking suggestion to apply it to AdamW. We do not apply post-clipping to AdamW since AdamW actually has an adaptive stepsize mechanism, and directly applying post-clipping will make the conclusion indistinct. The original Figure 4 shows the immediate effect of Newton-step clipping on K-FAC, and we supplement a set of results in Figure 4 of K-FAC + pre-clipping in the current revision,
>
> > Another interesting experiment would be to run AdamW longer and see if it can recover performance with Newton steps.
> - Your points are totally correct. AdamW can recover comparable testing performance to their second-order counterparts when trained long enough. But we would like to emphasize that for parameter-efficient training, convergence speed is more of a problem waiting for amelioration. Though remarkably freeing the computation and storage burden, parameter-efficient tuning generally lacks behind fine-tuning in terms of loss decay rate[1]. Hence, what we would like to devise is a setting that enables faster convergence within commonly adopted total epochs.
> - We agree that a set of experiments with a very long training process would be interesting. We supplement the study in Figure 11 of Appendix C.4, where we train the model on STS-B for 500 epochs with K-FAC + NewtonClip and AdamW. We observe that K-FAC could yield a smaller loss scale ($10^{-6}$) than AdamW ($10^{-4}$), and both methods do not suffer from over-fitting.
>
> > Written informally
> - Thanks for your advice. We have rewritten our sentences more formally in the new version.
> I was unable to find # of training steps for the experiments in the appendix. I think it is useful to report it.
> - Thanks for the suggestion, because different datasets have different numbers of instances, we report the number of training epochs in experiments to make the results fairer.  We now report the training steps in Appendix B.
>
> > The authors mention that "The search space for learning rate, Newton-step clipping scale, and damping factor are 1e − 2 ∼ 0.5, 0.1 ∼ 2.0 and 1e − 2 ∼ 1e − 6, respectively". To me, this is not specific enough. Please report either exact discrete points which were tested using a grid search or mention more details about an automatic tuner.
> - Thanks, we now report the exact discrete points in the revision, and we did use grid search in experiments. For AdamW, the search space of the learning rate and max norm are {1e-2, 1e-3, 1e-4, 5e-3} and {0.1, 1.0, 10}, respectively. For ours, the search space of learning rate, Newton-step clipping scale, and damping factor are {0.01, 0.05, 0.1, 0.5}, {0.1, 1.0, 1.5, 2.0}, and {1e-2, 1e-3, 1e-4, 1e-5, 1e-6}. We also supplement the ablation study of these hyper-parameters in Appendix C.1.

---

### Official Review · Reviewer_tTAh · 2022-11-01

**Confidence:** 4
**Clarity, Quality, Novelty And Reproducibility:** 1. Originality
- While the general id…
**Correctness:** 2
**Technical Novelty And Significance:** 2
**Empirical Novelty And Significance:** 2
**Recommendation:** 5

**Strength And Weaknesses:**

Strengths:
- The general idea of only applying second-order optimizers to adapter layers in Pre-Trained Models (PTMs) is interesting.
- The paper is generally clearly-written and well-motivated. The topic of the paper is relevant to the ICLR community.

Weaknesses:
- The justifications for gradient clipping (pre-clipping and post-clipping) are non-theoretical and weak. Moreover, the authors do not mention the existing works on damping and clipping, which I further address below.
- Algorithm 1 contains several undefined variables and functions, and it isn't easy to understand the exact algorithm.
- The empirical results in the paper are not convincing due to two reasons: (1) the authors do not include the results of (a) no clipping, (b) only pre-clipping, (c) only post-clipping, and (d) the proposed modification, and (2) the search space for hyperparameters such as learning rate seem to be much larger for K-FAC and I believe that the comparison to weakly tuned AdamW is unfair.

**Summary Of The Paper:**

The paper proposes using second-order optimizers (K-FAC) to train large-scale Pre-Trained Models (PTMs). The key idea is to only apply the second-order update on adapter layers to make the storage and computation of the gradient covariances in K-FAC feasible in large PTMs. To further stabilize the training, the authors introduce a clipping scheme to control the size of the K-FAC update. Empirically, the proposed method can adapt additional layers in PTMs with faster convergence and improved performance compared to the AdamW baseline.

**Summary Of The Review:**

While I believe that the key idea proposed in the paper is interesting, the paper has critical weaknesses in both technical and empirical aspects. At the moment, I recommend a score of 3 (reject).

---

> ### Author Response · Authors · 2022-11-11
> **Response to tTAh (3/3)**
>
> > Reproducibility
> - Thanks, we now submit the code in the supplementary material.
>
> > In Table 2, what are the previously published results? It is not referenced in the paper. Moreover, the results of the validation loss are not shown in Figure 5 (and Figure 6).
> - Sorry for the confusion, we have added the reference in the revision. The results are from [1].
> [1]. LoRA: Low-rank Adaptation of Large Language Models
>
> > The authors mention that K-FAC is more vulnerable to the choice of hyperparameters. It would helpful if the authors conduct an ablation study empirically showing this sensitivity.
> - Thanks for your suggestion, we supplement the ablation study of hyperparameters in Appendix C.1. For a relatively larger dataset (STS-B), the effect of hyper-parameters is not significant, while the model becomes more sensitive when dealing with a smaller dataset (RTE).
>
>
> > In the first line of the introduction, “Pre-trained models (PTMs)” → “Pre-Trained Models (PTMs).” Furthermore, I believe that the writing in the introduction can be improved.
> - Thanks for your suggestion. We have fixed the term and modified the introduction to more highlight the meaning of this application both on the PTM side and the second-order optimization side.
>
> > There needs to be a mention that Hessian is assumed to be invertible (strictly convex) in Eqn. 2.
> - Thanks for your suggestion. We have added this assumption in the new version.
>
> > After Eqn. 7, the authors mention that the tunable approach accounts for roughly 0.5% ~ 8% of the parameters, but I believe that this linearly scales with the bottleneck dimension. By choosing higher r, couldn’t we bound the additional parameters? My comment also applies to Eqn. 8.
> - Sorry for the confusion, this sentence means that by choosing different bottleneck dimensions, the number of tunable parameters is empirically from 0.5% to 8%. We certainly can bound the additional parameters by choosing r. We study the number of tunable parameters in Section 5.4 (in the original paper, it is Section 5.3).
>
> > For the last paragraph in Section 3, are the authors claims’ based on the KFAC or the full Hessian matrix?
> - Our claim of numerical tractability is based on a general class of second-order methods (train with PET) that divide the full Hessian (or Fisher) matrix into layer-dependent blocks. The detailed numerical results are listed with K-FAC as an example.
>
> > In Figure 2, why is there a fluctuation in memory usage? Is it due to the memory stored for a forward and a backward pass?
> - During the training process, there will be many intermediate results that need to be calculated, which will occupy considerable video memory, and when these results are released, the video memory occupation will be reduced. It is worth noting that these intermediate calculations do not need to occur in the backpropagation, but the intermediate results obtained in the forward propagation also cause relatively high peaks.
>
> > The e notation in the Appendix seems weird to me.
> - Sorry for the confusion, we modified it by removing the equation environment.  It stands for exponential in scientific notation.

---

> > ### Comment · Reviewer_tTAh · 2022-11-16
> > **Response to Authors**
> >
> > I thank the authors for their detailed reply. I acknowledge that I read the authors' responses and other reviews.
> >
> > First, the authors improved the presentation and cited related works regarding damping and KL clipping during the discussion. However, as reviewers xAVz and fdsh also pointed out, my initial concerns on the justification remain. It still needs to be clarified why the combination of pre and post clippings is required for these adapter tasks. Moreover, I have several additional comments and questions:
> >
> > - While the authors say, "... This approach has previously appeared as an exclusive skill for K-FAC training (Ba et al., 2016), but it would hopefully work for a wider range of second-order optimizers", the experiments in Section 5 exclusively studies K-FAC and does not support this claim.
> > - In Figure 4, are the hyperparameters (e.g., learning rate and damping) tuned independently for runs without and with clipping?
> > - (Minor) In Appendix C.2, "We investigate how the proposed clipping technique affects the raining procedure" --> "We investigate how the proposed clipping technique affects the training procedure".

---

> > > ### Author Response · Authors · 2022-11-18
> > > **Further Response to Reviewer tTAh**
> > >
> > > - Thanks for your reply! To clarify, we didn't combine the pre-clipping and post-clipping. We validate through experiments a great distinction between pre-clipping and post-clipping. Meanwhile, we present the notable superiority of post-clipping over its non-clipping and pre-clipping counterparts in terms of convergence rate and training stability.
> > > - Sorry for the unclearness. We did preliminary explorations on several second-order optimizers before systemic experiments in Section 5. And post-clipping could also stabilize the training for them (however they are still less stable than K-FAC). We exclusively study K-FAC in the paper because it is the most powerful and stable one among them. We have uploaded an experimental example of PCA (a second-order optimization method [1]) with RoBERTa-large+ Adapter on STS-B to this [anonymous link](https://imgur.com/a/iOqcZsN), the figure illustrates the training curve with and without post clipping.
> > >
> > >
> > > > In Figure 4, are the hyperparameters (e.g., learning rate and damping) tuned independently for runs without and with clipping?
> > > - In the comparison between clipping and non-clipping versions, we first tune the hyper-parameters for non-clipping runs and choose a stable one with the best performance. Then we directly use the identical hyper-parameter setting on their non-clipping counterparts, which verifies the necessity and robustness of the post-clipping method.
> > >
> > > > (Minor) In Appendix C.2, "We investigate how the proposed clipping technique affects the raining procedure" --> "We investigate how the proposed clipping technique affects the training procedure".
> > > - Fixed. Thanks!
> > >
> > >
> > > [1] A Class of Short-term Recurrence Anderson Mixing Methods and Their Applications. ICLR 2021.

---

> ### Author Response · Authors · 2022-11-11
> **Response to tTAh (2/3)**
>
>
> > The authors, after Eqn. 6, mention that they do not distinguish Newton’s method and natural gradient descent. However, I feel that this distinction is important and the paper needs correctly distinguish between Newton's update and natural gradient descent. Moreover, do the authors intend to say ∇θ2L(θ)≈F^(θ)?
>
> > As opposed to the authors’ claims, K-FAC can compute the gradient covariances from samples from predictive distribution and can be seen as an approximation to the true Fisher instead of empirical Fisher. To be more specific, Gi should not be the covariance of the actual gradients seen during training. These are important distinctions in second-order methods [1] and must be carefully described and implemented.
> - We totally agree with your point that Newton's method and natural gradient descent (NGD) are different. But within the range of our analysis, these two methods can be unified in the sense that both of them precondition the gradient to yield an update. Hence the pre-or-post clipping works in a similar way for these two classes of second-order optimization.
> - Sorry for the confusing notation. We agree that when approximating the FIM, K-FAC does not make use of the actual gradients, but the gradients with respect to output $s_i$, which is already clarified in Appendix A2. However, in Chapter 4, we are not discussing K-FAC, but general NGD (or Newton's method), which takes $u=F^{-1}g$  (or $u=H^{-1}g$) as an update. We are not talking about any specific evaluation procedure of FIM, but the fact that small gradients do not correspond to small updates. We happen to use the same notation G_i as the expression of K-FAC in Appendix A2, which has led to confusion. We have corrected the notations in our new version.
>
> > The justifications for gradient clipping (pre-clipping and post-clipping) are non-theoretical and weak. Moreover, the authors do not mention the existing works on damping and clipping, which I further address below.
> - We apologize for missing theoretical proof. It is in part due to our application-oriented mind. As mentioned in the general response, our primary goal is to unprecedentedly make second-order optimization tractable on billion-scale models and indicate some indispensable training tricks. Meanwhile, we find that the theoretical proof is hard to start with, since current related theories rely on strong assumptions which have not ever been verified on deep transformers.
> - Damping and clipping. Thanks for this important notice. We have added references to the two mentioned works. First, for damping. The aforementioned two works clearly address the motivation of damping. But in our work, we examine the actual performance of K-FAC with damping on deep transformers, and we indicate that the training result is highly sensitive to the choice of damping parameter. Second, for clipping. Ba et al. 's work has already included a one-line comment about clipping the updates, and it is indeed our mistake not to discover it. We explore update clipping from a broader horizon, not regarding it as an exclusive strategy affiliated to K-FAC. We explain why it is preferred compared to the vanilla gradient clipping, by general second-order optimization, both Fisher-preconditioned natural gradient descent and hessian-based methods.

---

> ### Author Response · Authors · 2022-11-11
> **Response to Reviewer tTAh (1/3)**
>
> Thanks for your constructive comments.
>
> > Algorithm 1 contains several undefined variables and functions, and it isn't easy to understand the exact algorithm.
> - Sorry for some of the confusing notation. We have added the definition of those previously undefined quantities in the new version. Briefly speaking, $\mathbf{h}_t$ stands for a group of intermediate quantities that are required for calculating the final update tensor. (We remove $\mathbf{r}_t$ for better clearness in the new version.) For example, in AdamW, $\mathbf{h}_t$ can be the concatenation of momentum and squared momentum; in K-FAC, $\mathbf{h}_t$ can be the approximate FIM. $\mathcal{T}$ stands for the transform from gradient to final update.
>
> > The results of (a) no clipping, (b) only pre-clipping, (c) only post-clipping, and (d) the proposed modification
> - Thanks. We have clarified in Section 4.3 that our Newton-step clipping is a kind of post-clipping, that is, it clips the gradient only after preconditioning.
> - The results with and without Newton-step clipping on K-FAC are shown in Figure 4. It may not be easy to notice because we put them in Section 4 but not in Section 5.  Now we have moved the results to Section 5.1, and they are still in Figure 4. We also supplement the results of K-FAC + pre-clipping in Figure 4 in the current revision. Also, an analytic experiment in Appendix C.2 (Figure 9) could also demonstrate its effectiveness.
>
> > The search space for hyperparameters such as learning rate seems to be much larger for K-FAC, and I believe that the comparison to weakly tuned AdamW is unfair.
> - Thanks, we now report the exact discrete points in the revision, and we did use grid search in experiments. For AdamW, the search space of the learning rate and max norm are {1e-2, 1e-3, 1e-4, 5e-3} and {0.1, 1.0, 10}, respectively. For ours, the search space of learning rate, Newton-step clipping scale, and damping factor are {0.01, 0.05, 0.1, 0.5}, {0.1, 1.0, 1.5, 2.0}, and {1e-2, 1e-3, 1e-4, 1e-5, 1e-6}. The size of the learning rate space of K-FAC is the same as AdamW, and the overall space is larger than AdamW.
> - But we do not think "unfair" is an appropriate term to describe the situation. This is because optimizing deep transformers with AdamW is extensively practiced in the community, and we can easily know and adopt the practical configurations. For example, according to our empirical experience, 1e-4 is an optimal or near-optimal learning rate for most scenarios to adapt deep transformers. However, we have no previous empirical evidence to guide the 2nd-order optimization on such large models. Thus, we use a comparatively larger search space and truthfully report it. On the other hand, it is also true that the second-order optimizer has more hyperparameters and that the damping factor has an impact on the optimization.

---

### Author Response · Authors · 2022-11-11
**General Response**

Dear reviewers, thanks very much for your comments. After carefully reading the reviews, we find they are exceedingly constructive and valuable for us to improve the work. Although the overall comments lean to negative, we are glad to see that the general idea is acknowledged to some extent. This work is challenging because we do not have much empirical or theoretical evidence to guide us to apply sophisticated 2nd-order optimization to such large models. Most of the previous work on second-order optimization did not attempt (nor was it computationally affordable) to experiment on large-scale models with hundreds of millions to billions of parameters. But we believe the application is important and meaningful as we combine two previously almost non-intersect settings for the first time. Specifically:
- On the model side, using second-order optimization may inspire better training methods and related analyses of giant language models.
- On the optimization side, we can evaluate the characteristics and practicability of previous 2nd-order optimizers under broader scenarios.
For example, in Figure 4, we find that directly using K-FAC to deep transformers in this setting cannot satisfactorily converge at all.

It would be encouraging if subsequent studies on 2nd-order optimization could conduct experiments on deep transformers with millions to billions of parameters rather than small-size networks. We will steadfastly improve the work, in the current revision:
- We add an ablation study of the hyper-parameters in Appendix C.1.
- We add an analysis of time efficiency in Appendix C.3.
- We add an analysis of the clipping occurrence in the training process in Figure 10 of Appendix C.2.
- We add experiments in Figure 9 of Appendix C.2 to illustrate that NewtonClip is not equivalent to turning down the learning rate.
- We add more experimental details in Appendix B.
- We add the results of validation metrics in Figure 8.
- We add a study of increasing the number of training epochs in Appendix C.4.
- We add the results of K-FAC + pre-clipping in Figure 4, note that Newton-step clipping is a type of post-clipping.
- We move Figure 4 (which directly shows the effectiveness of Newton-step clipping) from Section 4 to Section 5 in case some reviewers don't notice the results.
- We add detailed definitions to Algorithm 1 to make it more interpretable.
- We replace some notations in Section 4.2 to avoid potential confusion.
- We add two missing references and briefly discuss the connection between these works and ours.
- We submit our code in supplementary materials.

---

> ### Author Response · Authors · 2022-11-15
> **Follow-up General Response**
>
> Dear reviewers, we have uploaded the revision with major modifications and posted responses to your comments. Since the deadline for submitting revisions is approaching, we would appreciate it if you could take a look at the current revision and responses. We will also be pleased to have further discussions regarding the next-step improvement for the paper, thanks!

---

### Decision · Program_Chairs · 2023-01-20

**Decision:**

Reject

**Justification For Why Not Higher Score:**

See above.

**Justification For Why Not Lower Score:**

N/A

**Metareview: Summary, Strengths And Weaknesses:**

This paper proposes to train large models by combining two previously proposed ideas: K-FAC optimization and parameter-efficient tuning. In addition, it proposes to replace (or combine) gradient clipping with clipping of the final update.

On one hand, this paper is a nice demonstration that second-order optimization techniques can scale to 3-billion-parameter models. On the other hand, the reviewers feel (and I agree) that the novelty is fairly limited, as the proposed method combines two existing ideas in a relatively straightforward way. The clipping trick is fairly incremental, and various reviewers pointed out prior occurrences in the literature. The question, then, is whether the experimental results are compelling enough by themselves to justify acceptance; i.e., should they persuade us that the combination of PBT and K-FAC is more powerful than we might have naively supposed?  There was a fair amount of discussion between reviewers and authors about the details of the experimental validation, but at the end my impression is that the evidence is ambiguous. I hope the authors take the reviewers' feedback into account for resubmission.